# Snail determines the therapeutic response to mTOR kinase inhibitors by transcriptional repression of 4E-BP1

Jun Wang[1,2], Qing Ye[1,2], Yanan Cao[1,2], Yubin Guo[1,3], Xiuping Huang[1,3], Wenting Mi[1,3], Side Liu[3], Chi Wang[1,4], Hsin-Sheng Yang[1,5], Binhua P. Zhou[1,6], B. Mark Evers[1,7] & Qing-Bai She [1,2]

Loss of 4E-BP1 expression has been linked to cancer progression and resistance to mTOR inhibitors, but the mechanism underlying 4E-BP1 downregulation in tumors remains unclear. Here we identify Snail as a strong transcriptional repressor of 4E-BP1. We find that 4E-BP1 expression inversely correlates with Snail level in cancer cell lines and clinical specimens. Snail binds to three E-boxes present in the human *4E-BP1* promoter to repress transcription of *4E-BP1*. Ectopic expression of Snail in cancer cell lines lacking Snail profoundly represses 4E-BP1 expression, promotes cap-dependent translation in polysomes, and reduces the anti-proliferative effect of mTOR kinase inhibitors. Conversely, genetic and pharmacological inhibition of Snail function restores 4E-BP1 expression and sensitizes cancer cells to mTOR kinase inhibitors by enhancing 4E-BP1-mediated translation-repressive effect on cell proliferation and tumor growth. Our study reveals a critical Snail-4E-BP1 signaling axis in tumorigenesis, and provides a rationale for targeting Snail to improve mTOR-targeted therapies.

[1] Markey Cancer Center, University of Kentucky College of Medicine, Lexington, KY 40506, USA. [2] Department of Pharmacology and Nutritional Sciences, University of Kentucky College of Medicine, Lexington, KY 40506, USA. [3] Guangdong Provincial Key Laboratory of Gastroenterology, Department of Gastroenterology, Nanfang Hospital, Southern Medical University, Guangzhou, Guangdong 510515, China. [4] Department of Biostatistics, University of Kentucky College of Public Health, Lexington, KY 40506, USA. [5] Department of Toxicology and Cancer Biology, University of Kentucky College of Medicine, Lexington, KY 40506, USA. [6] Department of Molecular and Cellular Biochemistry, University of Kentucky College of Medicine, Lexington, KY 40506, USA. [7] Department of Surgery, University of Kentucky College of Medicine, Lexington, KY 40506, USA. Jun Wang, Qing Ye and Yanan Cao contributed equally to this work. Correspondence and requests for materials should be addressed to Q.-B.S. (email: qing-bai.she@uky.edu)

The kinase, mammalian target of rapamycin (mTOR), controls cell proliferation, survival, and metabolism by integrating a variety of signals from growth factors and nutrients[1]. The multiple functions of mTOR are exerted by the formation of two distinct protein complexes, mTOR complex 1 (mTORC1) and mTOR complex 2 (mTORC2). mTORC1 regulates messenger RNA translation to promote protein synthesis and cell proliferation by phosphorylating two primary downstream effectors, eukaryotic initiation factor 4E (eIF4E)-binding protein 1 (4E-BP1) and 70 kDa ribosomal protein S6 kinase 1. 4E-BP1 is a translational repressor, which prevents the assembly of the eIF4F translation initiation complex by competing with eIF4G for binding to eIF4E, a 5′ mRNA cap-binding subunit of the eIF4F complex[2]. Phosphorylation of 4E-BP1 by mTORC1 results in the release of eIF4E, thus increasing eIF4E availability for cap-dependent mRNA translation[3–5]. The mTORC2 complex predominantly controls cell survival and cytoskeleton organization by phosphorylating AKT and paxillin, among other proteins. mTOR signaling is activated in the majority of cancers owing to mutations in upstream signaling components including RAS, PI3K, PTEN, TSC1, TSC2, and LKB1[1]. Thus, mTOR signaling is believed to be an essential component for tumor development and progression, and targeting mTOR is thought to be a promising strategy for cancer therapy.

Rapamycin and related 'rapalogs' are allosteric inhibitors of mTORC1, and were among the first mTOR-targeted therapeutics in the treatment of solid tumors[6]. Rapalogs have shown some success in specific tumor types and in patients with rare somatic TSC mutations[7], but their overall activity as a monotherapy is limited[6]. This poor response might result from the weak inhibition of 4E-BP1 phosphorylation by rapalogs and induction of AKT activation through loss of the mTORC1/S6K-dependent negative feedback loops[1,6,8]. To overcome these issues, several new ATP-competitive mTOR kinase inhibitors (mTORkis) such as AZD8055[9] and INK128[10], which inhibit both mTORC1 and

mTORC2, have been developed and are being evaluated in clinical trials. Many of these small-molecule mTORkis profoundly abrogate phosphorylation of both 4E-BP1 and AKT, and exhibit superior anti-proliferative and anticancer activities compared with rapamycin. Nonetheless, a number of recent studies, including our own, demonstrate that deregulation of cap-dependent translation by an incomplete inhibition of 4E-BP1 phosphorylation, reduction of 4E-BP1 expression, or upregulation of eIF4E expression renders cancer cell resistance to mTORkis[11–16].

A growing body of evidence indicates that 4E-BP1 is a critical effector of mTOR signaling through translational control of key oncogenic mRNAs that encode proteins for cell-cycle progression, cell survival, angiogenesis, cancer progression, and metastasis[17]. 4E-BP1 expression is regulated by both transcriptional and posttranslational mechanisms[17]. Reduction of 4E-BP1 expression in cancer has been linked to malignant progression and poor prognosis[13,18]. However, the molecular mechanism underlying the decreased 4E-BP1 levels in cancer is poorly understood. We recently showed that 4E-BP1 negatively regulates cap-dependent translation of Snail, an important transcription factor triggering epithelial–mesenchymal transition (EMT) and promoting cancer invasion and metastasis[14]. Here we demonstrate that Snail acts as a reciprocal feedback suppressor of 4E-BP1 expression by blocking the transcription of the *4E-BP1* gene, which mitigates the antitumor activities of mTORkis in cancer cells with overexpression of Snail.

## Results

**4E-BP1 and Snail levels inversely correlate in cancer.** In a previous study[14], we uncovered the interesting fact that knockdown of Snail by small interfering RNAs (siRNAs) largely increases 4E-BP1 expression in HCT116 colon cancer cells. As Snail is a well-known transcriptional repressor capable of binding

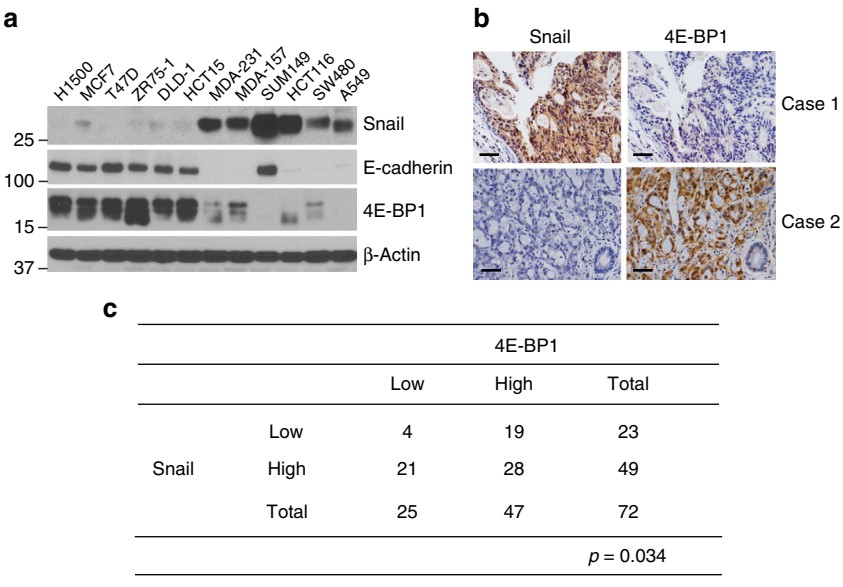

**Fig. 1** 4E-BP1 expression inversely correlates with Snail expression in cancer. **a** Cell lysates were prepared from the indicated cancer cell lines and analyzed by western blotting for the indicated proteins. **b, c** A tissue array of 72 human colorectal cancer specimens was subjected to immunohistochemical staining using antibodies against Snail (**b**, left) and 4E-BP1 (**b**, right). The immunoreactivity was scored blindly according to the value of immunoreaction intensity (none = 0; weak = 1; intermediate = 2; and strong = 3) and the percentage of tumor cell staining (none = 0; < 10 = 1; 10–50 = 2; > 50% = 3). The intensity and percentage values were added to provide a final immunoreactivity score ranging from 0 to 6. A score from 0 to 4 was defined as low protein expression and a score of 5–6 indicates a high level of protein expression. Representative staining images are shown in **b** and the immunoreactivity scores of the 72 specimens are summarized in a grid **c** that associates expression for both Snail and 4E-BP1. Statistical significance of the correlation between Snail and 4E-BP1 expression was determine by the $\chi^2$-test with the *p*-value indicated. Scale bar, 50 μm

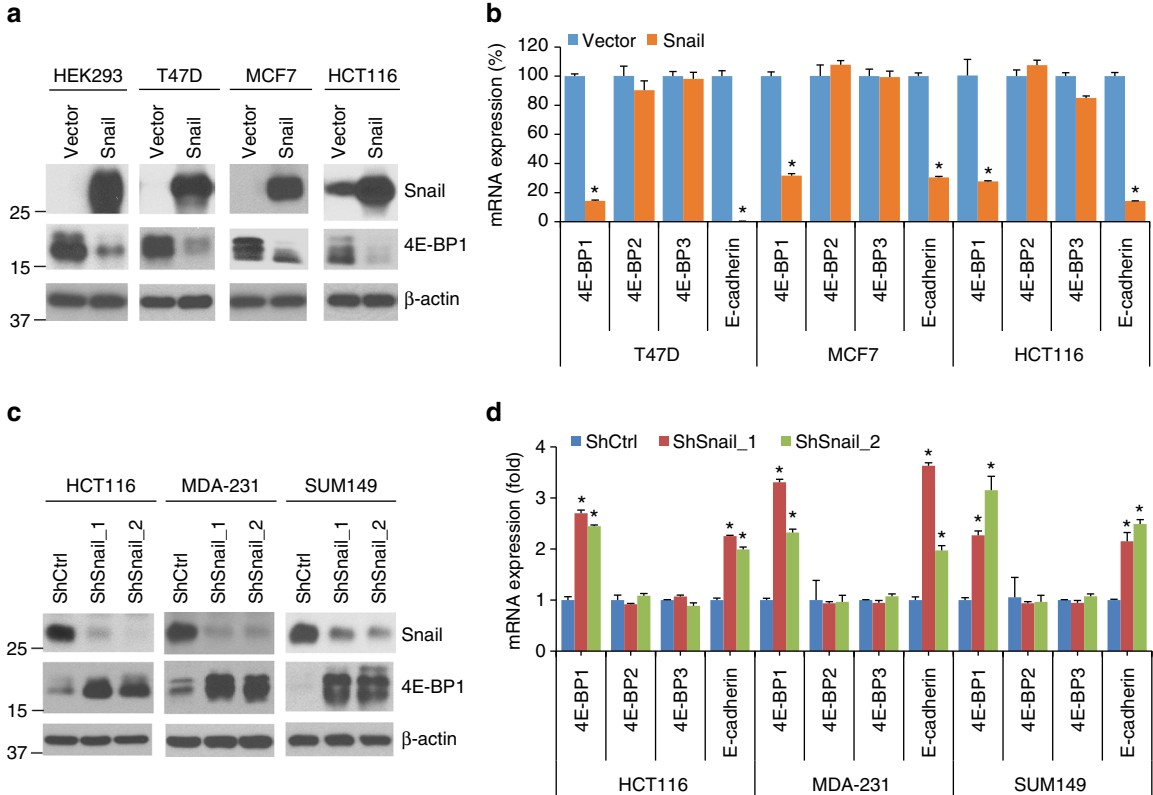

**Fig. 2** Snail represses 4E-BP1 expression at both the protein and mRNA levels. **a** HEK293, T47D, MCF7, and HCT116 cells with stable expression of Snail or vector control were analyzed by western blotting for the indicated proteins. **b** mRNA expression of the indicated genes was analyzed by quantitative RT-PCR in T47D, MCF7, and HCT116 cells with stable expression of Snail or vector control. The indicated gene expression was normalized against GAPDH and presented as a percentage of the expression level found in vector control cells. **c** HCT116, MDA-231, and SUM149 cells with stable expression of two different sets of Snail shRNAs (ShSnail_1 and ShSnail_2) or control shRNA (ShCtrl) were analyzed by western blotting for the indicated proteins. **d** mRNA expression of the indicated genes was analyzed by quantitative RT-PCR in HCT116, MDA-231, and SUM149 cells with stable expression of ShSnail_1, ShSnail_2, or ShCtrl. The indicated gene expression was normalized against GAPDH and presented as a fold increase over the expression level found in ShCtrl cells. All graphic data are presented as mean ± SEM ($n = 3$ technical replicates per condition). *$P < 0.001$ for Snail vs. vector or ShSnail vs. ShCtrl using Student's $t$-test

and inhibiting promoter activity of many target genes such as *E-cadherin* and phosphatase and tension homolog (*PTEN*)[19–21], we hypothesized that 4E-BP1 may be a transcriptional target of Snail. We first performed western blot analysis on a panel of breast (H1500, MCF7, T47D, ZR75-1, MDA-231, MDA-157, and SUM149), colon (DLD-1, HCT15, HCT116, and SW480), and lung (A549) cancer cell lines. The results of this analysis could be split into two groups: low and high Snail protein expression levels, and we found that Snail protein levels inversely correlated with expression of 4E-BP1 (Fig. 1a). Consistent with previous studies[19], Snail expression was also inversely associated with E-cadherin expression in all except the SUM149 cell line. Of note, as compared with E-cadherin expression, 4E-BP1 expression was not detectable in the Snail-overexpressing SUM 149 cells (Fig. 1a). Using immunohistochemical staining, we expanded the examination of Snail and 4E-BP1 expressions in colorectal cancer clinical specimens, and a significant inverse correlation was found (Fig. 1b, c).

**Snail selectively downregulates 4E-BP1 expression.** The inverse relationship between Snail and 4E-BP1 protein levels prompted us to investigate the ability of Snail to regulate 4E-BP1 expression. First, we expressed Snail in three cell lines that lack Snail protein (HEK293, T47D, and MCF7). Ectopic expression of human Snail almost completely suppressed 4E-BP1 protein expression in these cell lines (Fig. 2a). Moreover, overexpression of Snail in HCT116 cells, which endogenously expresses Snail, further repressed 4E-

BP1 expression (Fig. 2a). Similar to the well-characterized repression of *E-cadherin* mRNA expression by Snail, Snail-expressing cells (T47D, MCF7, and HCT116) also showed a dramatic reduction of *4E-BP1* mRNA expression (Fig. 2b). To determine whether 4E-BP family members, 4E-BP2 and 4E-BP3, are also regulated by Snail, we designed specific primer sequences to selectively determine their mRNA expression. Interestingly, the mRNA level between Snail-expressing and control cells for *4E-BP2* or *4E-BP3* was not changed (Fig. 2b). On the other hand, knockdown of Snail with stable expression of two different sets of short hairpin RNAs (shRNAs) in three cancer cell lines expressing high levels of Snail (HCT116, MDA-231, and SUM149) resulted in a profound induction of 4E-BP1 expression at both the protein and mRNA levels (Fig. 2c, d). *E-cadherin* mRNA expression was also markedly upregulated, but the levels of 4E-BP2 and 4E-BP3 remained unchanged in response to Snail knockdown. Collectively, these data reveal that Snail selectively downregulates *4E-BP1* gene expression.

To further validate the role of Snail in the regulation of 4E-BP1 expression, we generated *Snail* knockout (KO) HCT116 and MDA-231 cells using the CRISPR-Cas9 nickase system[22]. Sequencing confirmed that two types of frameshift indels were created in the targeted region of *Snail* exon 1 in the KO cells, but not in the wild-type (WT) cells (Supplementary Fig. 1a). In both HCT116 and MDA-231 cell lines, disruption of *Snail* markedly increased 4E-BP1 expression (Supplementary Fig. 1b). Importantly, re-expression of Snail in the two *Snail* KO-HCT116 or

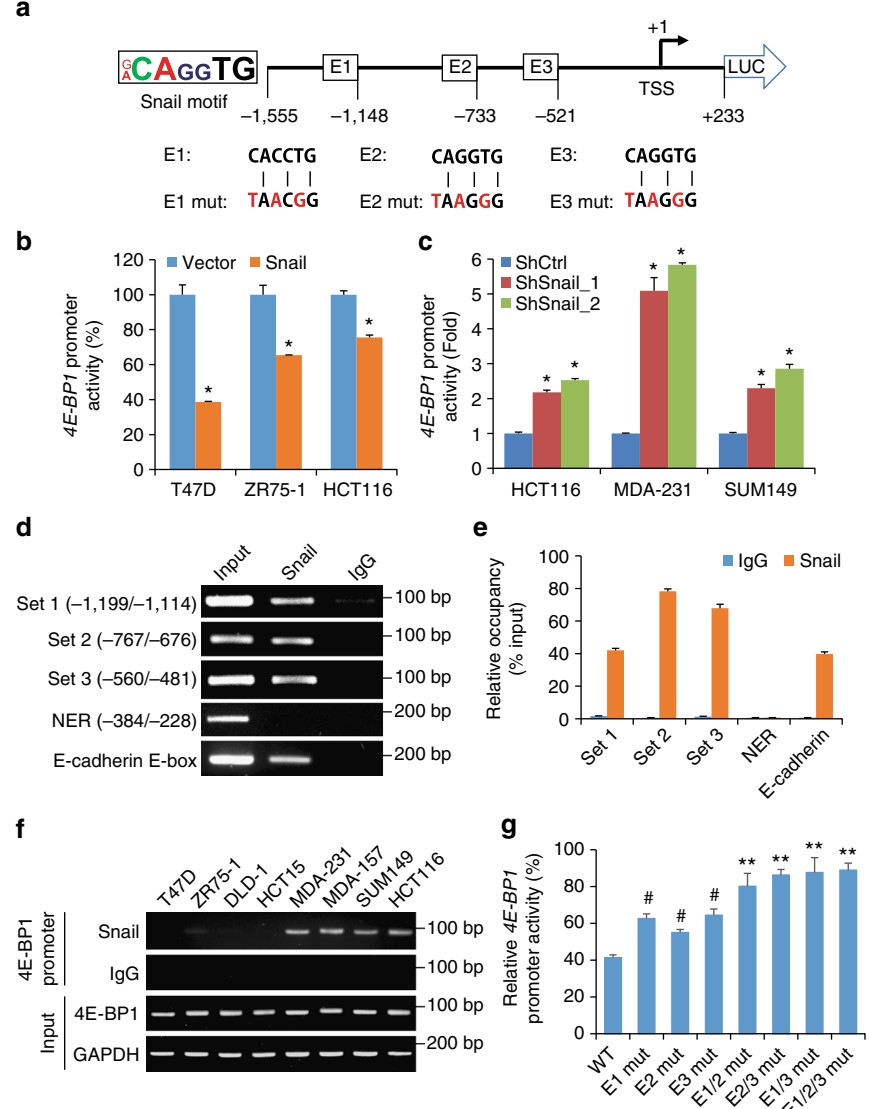

**Fig. 3** Snail represses *4E-BP1* promoter activity. **a** Schematic diagram of the human *4E-BP1* promoter-firefly luciferase construct (4E-BP1-Luc) showing positions of the three potential Snail-binding E-boxes (E1, E2, and E3). Mutated (Mut) nucleotides in each E-box sequence are indicated with red characters. **b**, **c** The indicated cell lines with expression **b** or knockdown **c** of Snail or their respective controls were co-transfected with human 4E-BP1-Luc and an internal control *Renilla*-Luc. After 36 h, luciferase activities were measured and normalized. The results are expressed as a percentage of the activity found in vector control cells **b** or as a fold increase over the Luc activity found in shRNA control (ShCtrl) cells **c**. **d** ChIP analysis of Snail-expressing HEK293 cell extracts using a specific antibody against Snail or an irrelevant IgG. Four sets of primers cover E1, E2, and E3 boxes and non-E-box region (NER) on the *4E-BP1* promoter, and a set of primers covers an E-box of the *E-cadherin* promoter as a positive control. **e** Enrichment levels for Snail or an irrelevant IgG at four regions of *4E-BP1* promoter and an E-box region of *E-cadherin* promoter as shown in **d** are presented as a percentage of input. **f** ChIP analysis of DNA from different cell lines using a specific antibody against Snail and the set 3 primers described in **d**. **g** Snail-expressing or vector control T47D cells were co-transfected with pGL3-4E-BP1-Luc (WT, wild type) or the same plasmid containing mutations in different E-boxes together with an internal control *Renilla*-Luc. Data are expressed as Luc activity in the presence of exogenous Snail as a percentage of the activity of the same reporter in vector control cells. All graphic data are presented as mean ± SEM ($n = 3$ technical replicates per condition). *$P < 0.001$ for Snail vs. vector or ShSnail vs. ShCtrl; #$P < 0.001$ for each mutated E-box vs. WT; **$P < 0.02$ for the combination of mutated E-boxes vs. WT or each mutated E-box. Statistical significance was determine by Student's *t*-test

MDA-231 cell clones restored the ability of Snail to repress 4E-BP1 expression (Supplementary Fig. 1c). Snail is highly expressed in fibroblasts in association with loss of E-cadherin expression[23]. Interestingly, silencing Snail using siRNAs in two Snail-expressing normal human fetal lung fibroblasts (IMR-90 and TIG1) also dramatically increased the expression levels of both 4E-BP1 and E-cadherin (Supplementary Fig. 2). Thus, these results corroborate that Snail is a critical repressor of 4E-BP1 expression.

**Snail directly represses *4E-BP1* promoter activity**. To explore the molecular mechanism by which Snail could repress the transcription of *4E-BP1*, we first analyzed the human *4E-BP1* genomic sequence and found that the *4E-BP1* promoter contains three putative Snail-binding E-boxes[24] (5′-CAGGTG-3′ or 5′-CACCTG-3′) upstream of its transcription start site (Fig. 3a and Supplementary Fig. 3a). We cloned a fragment of the human *4E-BP1* promoter (position − 1,555/+ 233) containing the three E-boxes and fused it to a firefly luciferase reporter. By transient

transfection with this *4E-BP1* promoter reporter into T47D, ZR75-1 and HCT116 cells that stably expressed either Snail or vector control, we found that Snail expression significantly repressed *4E-BP1* promoter activity in these cells (Fig. 3b). Conversely, silencing Snail using shRNAs in HCT116, MDA-231 and SUM149 cells or disruption of *Snail* in HCT116 and MDA-231 cells induced two to six fold increase in the *4E-BP1* promoter activity (Fig. 3c and Supplementary Fig. 3b). To determine

whether Snail binds to regulatory regions of the *4E-BP1* promoter, we performed chromatin immunoprecipitation (ChIP) analysis in HEK293 cells expressing Snail using three sets of primers; these covered E-box 1 (− 1,199/− 1,114), 2 (− 767/− 676), and 3 (− 560/− 481) sequences of the *4E-BP1* gene, respectively (Fig. 3d). Snail occupied all three E-box regions of the *4E-BP1* promoter (Fig. 3d), although the relative Snail occupancy was greater at E-boxes 2 and 3 than E-box 1 (Fig. 3e). In contrast,

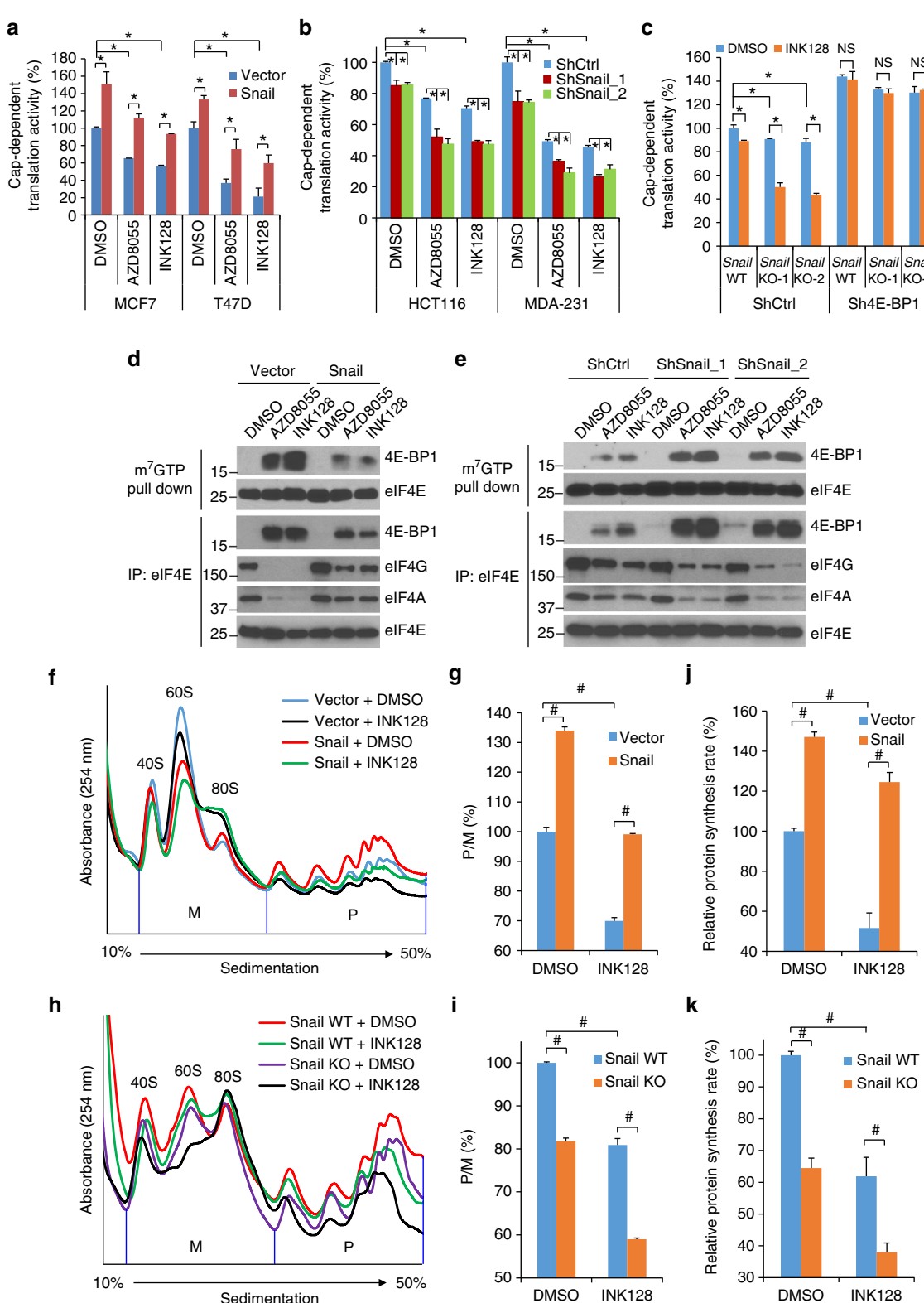

Snail did not bind to the non-E-box region (− 384/− 228) of the *4E-BP1* promoter, but did bind to the E-box region of the *E-cadherin* gene (Fig. 3d, e). Moreover, a subsequent ChIP experiment using a panel of cell lines showed endogenous Snail binding to the *4E-BP1* promoter in the cell lines that have high Snail/low 4E-BP1 expression, but not in cell lines with low Snail/high 4E-BP1 expression (Figs 1a and 3f). To further define the putative E-box elements inside the human *4E-BP1* promoter that are involved in repression by Snail, each E-box sequence was mutated (Fig. 3a). We compared the ability of Snail to repress different reporter constructs carrying each mutated E-box alone and in combination in Snail-expressing T47D cells. These experiments revealed that all three E-boxes were required for and cooperated in the Snail-mediated *4E-BP1* repression (Fig. 3g). These data suggest that Snail specifically and directly binds to the three E-boxes on the human *4E-BP1* promoter to repress its transcription.

To test whether the direct regulation of 4E-BP1 by Snail is conserved among mammals, we searched for the Snail-binding E-box of the *4E-BP1* promoter from different species. Interestingly, we found that the human Snail-binding E-boxes (5′-CAGGTG-3′ or 5′-CACCTG-3′) and their surrounding sequences are highly conserved among primate species, chimpanzee and monkey (Supplementary Fig. 4a). The mouse and rat do not share the exact human Snail-binding E-box sequences on their *4E-BP1* promoters, but the 5′-CACTTG-3′ or 5′-CAAGTG-3′ core, which is a relatively strong Snail-binding motif[24], was found on their *4E-BP1* promoters and is highly conserved between mouse and rat (Supplementary Fig. 4b). One E-box sequence (position − 75/− 70) was identified immediately upstream of the mouse *4E-BP1* transcriptional start site and two additional E-boxes were identified at positions at − 929/− 924 and − 940/− 935 from the transcription start site (Supplementary Fig. 4b). We tested the ability of Snail to repress the isolated mouse *4E-BP1* promoter (position − 1,430/+ 80) by luciferase reporter assay. Expression of either human or mouse Snail effectively repressed mouse *4E-BP1* promoter activity, as well as its expression at both the protein and mRNA levels in NMuMG mouse mammary epithelial cells and in 4T1 mouse mammary carcinoma cells (Supplementary Fig. 4c–e). Conversely, knockdown of mouse Snail expression in NIH3T3 fibroblasts markedly upregulated 4E-BP1 protein and mRNA expression (Supplementary Fig. 4f, g). As a positive control, mouse E-cadherin expression was also negatively regulated by Snail expression (Supplementary Fig. 4d–g). Similarly, the canine *4E-BP1* promoter also contains three putative Snail-binding E-boxes upstream of its transcription start site (Supplementary Fig. 5a). Ectopic expression of human Snail also profoundly suppressed 4E-BP1 promoter activity and its expression in Madin–Darby canine kidney (MDCK) cells (Supplementary Fig. 5b–d). Taken together, these results indicate that 4E-BP1 is a conserved transcriptional target of Snail.

Slug and Zeb1 are two transcriptional factors that mimic Snail repression of E-cadherin through binding to the same E-box elements[25,26], whereas the Twist1 transcriptional factor indirectly represses E-cadherin expression through binding to a different E-box motif[27]. To test whether these EMT-inducing transcriptional factors also regulate 4E-BP1 expression, Slug, Zeb1, and Twist1 genes were transiently transfected in T47D cells that lack expression of these transcriptional factors (Supplementary Fig. 6a). Similar to Snail expression, Slug expression strongly repressed 4E-BP1 promoter activity and its mRNA and protein expression, whereas expression of Zeb1 or Twist1 did not significantly inhibit 4E-BP1 expression but did markedly repress E-cadherin expression (Supplementary Fig. 6b–d). Importantly, knockdown of Slug in MDA-231 and MDA-157 breast cancer cell lines that express high levels of Slug, by either transient transfection with siRNAs or stable expression of specific shRNAs, provided results similar to the knockdown of Snail: a profound increase in 4E-BP1 expression at both the mRNA and protein levels (Supplementary Fig. 6a, e–j). Silencing Zeb1 provided a dramatic increase in E-cadherin expression but had no effect on 4E-BP1 expression in both cell lines (Supplementary Fig. 6e, f, i, j). Moreover, silencing Twist1 also showed no effect on 4E-BP1 expression but significantly upregulated E-cadherin expression in the Twist1-overexpressing MDA-157 cells (Supplementary Fig. 6a, i, j). These data suggest that the Snail family member, Slug, may also function as a strong transcriptional repressor of 4E-BP1.

**Snail modulates mTOR/4E-BP1-mediated translation.** Loss of 4E-BP1 expression activates cap-dependent translation and causes cancer cell resistance to mTORkis[12–14]. Therefore, we investigated whether Snail regulates cap-dependent translation by repressing 4E-BP1 and thereby alters the ability of mTORkis to effect translation. Using a dual luciferase reporter system that monitors the ratio between cap-dependent and -independent translation initiation[28,29], we found that expression of Snail in MCF7 and T47D cells significantly increased the cap-dependent translation rate but had no effect on initiation at internal ribosome entry sites (IRES)-driven cap-independent translation; in addition, Snail expression largely attenuated the cap-dependent translation inhibition induced by mTORkis, AZD8055, and INK128 (Fig. 4a and Supplementary Fig. 7a). Conversely, knockdown of Snail in HCT116 and MDA-231 cells repressed cap-dependent but not cap-independent translation, and the inhibitory effect on cap-dependent translation was enhanced by AZD8055 or INK128 (Fig. 4b and Supplementary Fig. 7b). Most importantly, with respect to cap-dependent translation, depletion of 4E-BP1 completely reversed the inhibitory effects of *Snail* knockdown or KO alone, and in combination with INK128 in HCT116 cells (Fig. 4c and Supplementary Fig. 7c, d), suggesting

**Fig. 4** Snail influences mTOR/4E-BP1-controlled translation and polysome formation. **a**, **b** Snail-expressing **a** or -knockdown **b** cells and their respective controls were transfected with a bicistronic luciferase reporter that detects cap-dependent translation of the *Renilla* luciferase gene and cap-independent translation of the firefly luciferase gene. After 24 h, cells were treated with mTOR kinase inhibitors AZD8055 (100 nM), INK128 (100 nM), or DMSO for 12 h, followed by measurement of luciferase activities. The ratio of *Renilla*/firefly luciferase activities was calculated and presented as a percentage of the cap-dependent translation activity found in the DMSO-treated control cells. **c** Cap-dependent translation analysis of *Snail* wild-type (WT) and knockout (KO) HCT116 cells expressing 4E-BP1 shRNA or control shRNA treated with 100 nM INK128 for 12 h. **d**, **e** Snail-expressing T47D cells **d** or Snail-knockdown HCT116 cells **e** and their respective controls were treated with 100 nM AZD8055, 100 nM INK128, or DMSO for 12 h. Cell lysates were precipitated with m7GTP sepharose beads or immunoprecipitated with eIF4E antibody followed by western blot analysis for the indicated proteins. **f–i** Snail-expressing or vector control MCF7 cells **f**, **g** and *Snail* WT or KO HCT116 cells **h**, **i** were treated with 100 nM INK128 or DMSO as control for 12 h, followed by polysome analysis. The vertical blue line separates the polysomal (P) and monosomal (M) fractions. The P/M ratio, an index of translational efficiency, was calculated by comparing areas under the polysome and monosome peaks using NIH image J. The results are expressed as a percentage of the P/M ratio relative to the DMSO-treated vector **g** or Snail WT **i** controls. **j**, **k** Nascent protein synthesis was measured for Snail-expressing or vector control T47D cells **j** and *Snail* WT or KO HCT116 cells **k** treated with 100 nM INK128 or DMSO for 12 h. The results are expressed as a percentage of methionine analog incorporation relative to the DMSO-treated control cells. All graphic data are presented as mean ± SEM ($n = 3$ technical replicates per condition). *$P < 0.01$; #$P < 0.001$; NS, not significant using Student's *t*-test

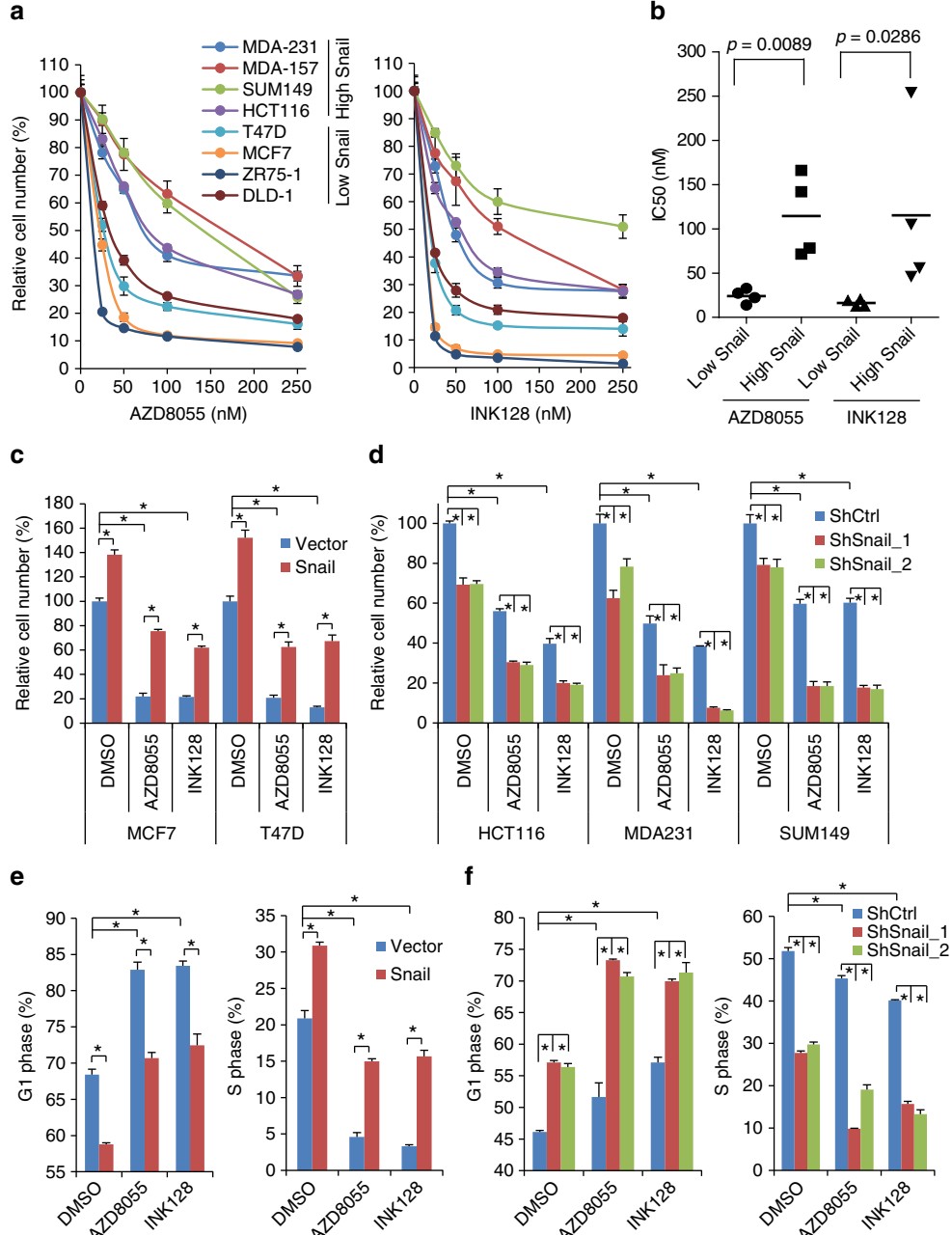

**Fig. 5** Snail mitigates the anti-proliferative effect of mTOR kinase inhibitors. **a**, **b** Growth for the indicated cell lines was assessed after 72 h of treatment with AZD8055 or INK128 (0–250 nM). Results are expressed as a percentage of cell number relative to the DMSO-treated control cells **a** or as the half-maximal growth inhibitory concentration (IC50) for AZD8055 or INK128 **b**. **c**, **d** The growth of the indicated cells with stable expression **c** or knockdown **d** of Snail or their respective controls was assessed after 72 h of treatment with 100 nM AZD8055, 100 nM INK128 or DMSO as control. The results are expressed as a percentage of cell number relative to the DMSO-treated vector control cells **c** or shRNA control cells **d**. **e**, **f** T47D cells with stable expression of Snail or vector control **e** and MDA-231 cells with stable expression of Snail shRNA or control shRNA **f** were treated with 100 nM AZD8055, 100 nM INK128 or DMSO as control for 24 h, followed by cell cycle analysis using flow cytometry. The results are expressed as the percentage of cells in the G1 or S phase distributions. All graphic data are presented as mean ± SEM ($n = 3$ technical replicates per condition). *$P < 0.001$ using Student's $t$-test

that Snail modulation of mTOR-regulated cap-dependent translation is mediated through 4E-BP1.

Although silencing Snail increased 4E-BP1 expression (Fig. 2c), it is likely to be that a large fraction of 4E-BP1 in cancer cells is hyperphosphorylated by oncogenic PI3K/AKT and RAS/ERK signaling[17,29]; the ability of 4E-BP1 to repress cap-dependent translation is compromised when in the phosphorylated state[3,5,17,29,30]. Dephosphorylated 4E-BP1 by mTOR inhibition competes with eIF4G to bind eIF4E and prevents formation of

eIF4F complex (eF4E, eIF4A, and eIF4G), thus inhibiting eIF4E-initiated cap-dependent translation[3,5,17]. To examine whether the effect of Snail expression on mTOR-regulated cap-dependent translation is associated with the level of 4E-BP1 bound to eIF4E, we first examined the eIF4E-mRNA cap-binding capacity of 4E-BP1 using m7GTP Sepharose beads, which mimics the 5'-mRNA cap to precipitate cap-interacting proteins. As shown in Fig. 4d, e Snail expression in T47D cells profoundly decreased the binding of 4E-BP1 to eIF4E induced by either AZD8055 or INK128,

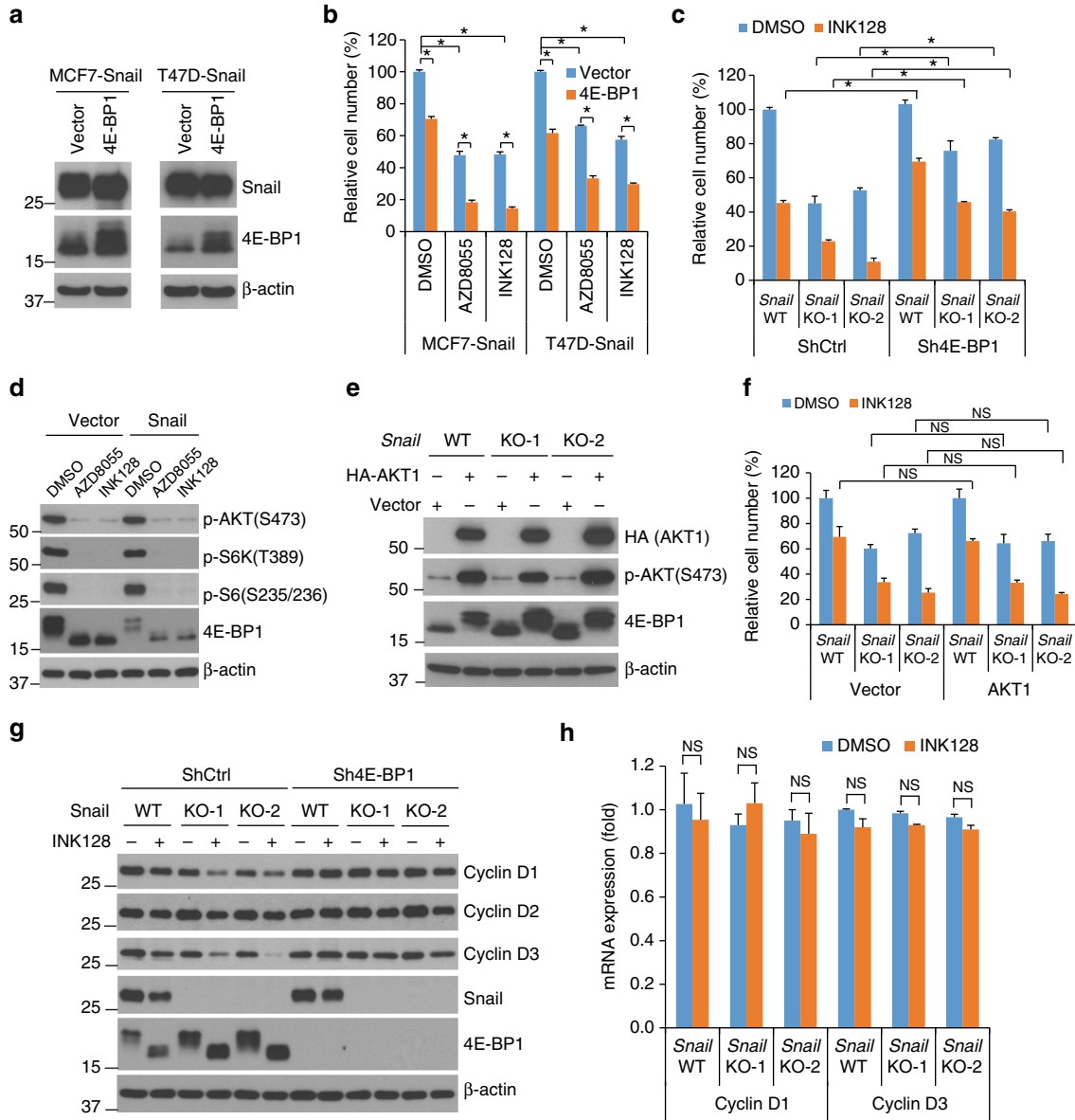

**Fig. 6** Snail modulation of cancer cell sensitivity to mTORkis is mediated at the level of 4E-BP1. **a, b** Snail-expressing MCF7 or T47D cells stably expressing exogenous 4E-BP1 or vector control were assessed by western blot analysis for the indicated proteins **a** or by cell growth analysis after 72 h of treatment with 100 nM AZD8055, 100 nM INK128, or DMSO as control **b**. **c** *Snail* WT or KO HCT116 cells stably expressing 4E-BP1 shRNA or control shRNA were assessed for cell growth after 72 h of treatment with 100 nM AZD8055, 100 nM INK128 or DMSO as control. The results are expressed as a percentage of cell number relative to the DMSO-treated *Snail* WT cells expressing control shRNA. **d** T47D cells with stable expression of Snail or vector control were treated with 100 nM AZD8055, 100 nM INK128, or DMSO as control for 12 h, followed by western blot analysis for the indicated proteins. **e, f** *Snail* WT or KO HCT116 cells expressing AKT1 or vector control were assessed by western blot analysis for the indicated proteins **e** or by cell growth analysis after 72 h of treatment with 100 nM INK128 or DMSO as control **f**. The results are expressed as a percentage of cell number relative to the DMSO-treated *Snail* WT cells expressing vector control. **g, h** *Snail* WT or KO HCT116 cells stably expressing 4E-BP1 shRNA or control shRNA were treated with 100 nM INK128 or DMSO as control for 12 h, followed by western blot analysis for the indicated proteins **g** or by quantitative RT-PCR analysis for mRNA expression of cyclin D1 and cyclin D3 relative to the levels found in DMSO-treated control cells **h**. All graphic data are presented as mean ± SEM (*n* = 3 technical replicates per condition). *$P < 0.001$; NS, not significant using Student's *t*-test

whereas silencing Snail in HCT116 cells had the opposite effect, with an increased 4E-BP1-eIF4E interaction when cells were treated with either mTORki. This observation in response to Snail expression or knockdown was confirmed by co-immunoprecipitation for the 4E-BP1-eIF4E interaction (Fig. 4d, e). Furthermore, the decreased 4E-BP1-eIF4E interaction in Snail-expressing T47D cells treated with mTORkis was accompanied by an increased formation of eIF4F complex as noted by increased levels of eIF4G and eIF4A bound to eIF4E (Fig. 4d). Conversely, the increased 4E-BP1-eIF4E interaction in Snail-depleted

HCT116 cells treated with mTORkis was accompanied by decreased formation of the eIF4F complex (Fig. 4e). Together, these data demonstrate that Snail-mediated repression of 4E-BP1 promotes cap-dependent translation and relieves the inhibitory effect of mTORkis on the translation.

mRNA is actively translated in polysomes and changes in overall translational efficiency are reflected by the changes in the ratio of polysomal (P) to monosomal (M) fractions[28,29]. Polysome profile analysis showed that Snail expression in MCF7 cells increased the P/M ratio (34%) compared with cells

expressing vector control (Fig. 4f, g). Treatment with INK128 decreased polysome assembly with a 30% reduction in the P/M ratio of MCF7-vector cells, but this reduction was almost completely abrogated by expression of Snail (Fig. 4f, g). By contrast, both *Snail* KO and treatment with INK128 reduced the P/M ratio (20–22%) in HCT116 cells (Fig. 4h, i). However, *Snail* KO in combination with INK128 further impaired polysome assembly by reducing the P/M ratio up to 41%. These effects were corroborated by measuring nascent protein synthesis (Fig. 4j, k). Overexpression of Snail in T47D cells increased protein synthesis and largely prevented the inhibitory effect of INK128 on protein synthesis (Fig. 4j), whereas *Snail* KO in combination with INK128 in HCT116 cells resulted in a more marked suppression of protein synthesis (62%) than either *Snail* KO or INK128 treatment alone (35–38%) (Fig. 4k). In total, these data indicate that Snail promotes recruitment of capping mRNAs to eIF4F for their active translation in polysomes, and that mTOR-regulated cap-dependent translation rate is altered by Snail-mediated repression of 4E-BP1.

**Snail reduces the anti-proliferative effect of mTORkis.** We next examined whether Snail confers an increased resistance to mTORkis through decreased 4E-BP1 expression. We compared the effect of mTORkis on growth in cancer cells with low and high levels of Snail. As compared with cancer cells with low Snail/high 4E-BP1 expression (MCF7, T47D, ZR75-1, and DLD-1, Fig. 1a), cancer cells expressing high Snail/low 4E-BP1 (MDA-231, MDA-157, SUM149, and HCT116, Fig. 1a) were significantly less sensitive to both AZD8055 and INK128, with a five to seven fold increase in half-maximal growth inhibitory concentration values (Fig. 5a, b). Notably, ectopic expression of Snail in MCF7 and T47D cells significantly increased cell growth (Fig. 5c). Treatment with either AZD8055 or INK128 profoundly inhibited growth of these two cell lines, but this suppression was largely alleviated by Snail expression (Fig. 5c). By contrast, silencing Snail using shRNAs in HCT116, MDA-231, and SUM149 cells inhibited their growth and markedly augmented the sensitivity of these cells to both AZD8055 and INK128 (Fig. 5d). Similar results were observed by disruption of *Snail* in HCT116 and MDA-231 cells (Supplementary Fig. 8a). A long-term colony formation assay that incorporates chronic treatment with AZD8055 or INK128 in Snail-depleted HCT116 cells confirmed the sensitization of these cells to mTORkis (Supplementary Fig. 8b). Cell cycle analysis revealed that Snail expression in T47D cells significantly promoted G1-S progression and alleviated the mTORkis-induced inhibition of G1-S progression, whereas silencing Snail in HCT116 cells suppressed G1-S progression and further enhanced inhibition of G1-S progression by mTORkis but did not exert a major effect on cell survival (Fig. 5e, f). These results imply that Snail alters the sensitivity of cancer cells to mTORkis largely through a regulation of cell cycle progression.

To evaluate whether the Snail-regulated 4E-BP1 level controls the sensitivity of cells to mTORkis, we first stably expressed exogenous 4E-BP1 to rescue the loss of 4E-BP1 in Snail-expressing MCF7 and T47D cells (Figs 2a and 6a). Interestingly, we found that restoration of 4E-BP1 suppressed growth in Snail-expressing MCF7 or T47D cells and sensitized these cells to AZD8055 or INK128 treatment (Fig. 6b). Next, silencing 4E-BP1 to reverse the increased 4E-BP1 level induced by *Snail* KO in HCT116 cells, largely attenuated cell growth inhibition by *Snail* KO alone or in combination with INK128 (Fig. 6c, g). These data reveal a critical role for 4E-BP1 in the modulation of cell sensitivity to mTORkis by Snail.

As Snail has been reported to activate AKT by transcriptional repression of PTEN[21], we tested whether PTEN/AKT signaling is also regulated by Snail and confers increased resistance to mTORkis in our tested cell lines. Among the three cell lines (T47D, MCF7, and HCT116) expressing exogenous Snail as shown in Fig. 2a, Snail expression only slightly repressed PTEN expression, coincident with enhanced activation of AKT in T47D cells but not in the two other cell lines (Supplementary Fig. 9a, b). Treatment with either AZD8055 or INK128, at the same concentration (100 nM) used to evaluate cell growth, effectively inhibited phosphorylation of mTOR substrates including AKT, S6K, and 4E-BP1, as well as the S6K substrate, S6 ribosomal protein, in both Snail-expressing or vector control T47D cells (Fig. 6d). Furthermore, Snail-expressing T47D cells with decreased 4E-BP1 level (Figs 2a and 6d) were significantly less sensitive to mTORkis compared with the vector control cells (Fig. 5c). In addition, silencing Snail in HCT116, MDA-231, and SUM149 cells (Fig. 2c) did not affect PTEN expression or AKT/S6K phosphorylation (Supplementary Fig. 9c,d). It is important to note that SUM149 cells have a naturally occurring PTEN loss associated with constitutive activation of AKT[31] (Supplementary Fig. 9c) and silencing Snail still sensitized SUM149 cells to mTORkis (Fig. 5d), suggesting that the increased resistance to mTORkis by Snail expression is independent of AKT activation. This notion was supported when overexpression of AKT1 in Snail-depleted HCT116 cells showed no significant change in growth inhibition by *Snail* KO alone or in combination with INK128 (Fig. 6e, f).

The G1-phase promoters, D-cyclins, are regulated by cap-dependent translation[12,32,33]. We found that the expression of cyclins D1 and D3 but not cyclin D2 was downregulated in *Snail* KO HCT116 cells (Fig. 6g). It has been shown that Snail can transcriptionally repress cyclin D2 expression in MDCK cells[34]. Consistent with this finding, we observed suppression of cyclin D2 and other Snail-repressed genes, including *E-cadherin* and *PTEN*, in MDCK cells when expressing Snail using our Snail construct[35] (Supplementary Fig. 5c, d). However, no change in cyclin D2 expression was found in Snail-expressing or -knockdown cell lines used in this study (Supplementary Fig. 9a, c). Treatment with INK128 in HCT116 cells downregulated Snail expression (Fig. 6g), which is likely suppressed by dephosphorylated 4E-BP1 in the cap-dependent translation level as reported previously[14]. However, downregulation of Snail did not result in a significant change in 4E-BP1 levels (Fig. 6g), which may have occurred because the levels of Snail bound to the 4E-BP1 promoter were not affected by INK128 treatment (Supplementary Fig. 10). Nevertheless, treatment of HCT116 cells with INK128 inhibited the expression of all D-cyclins, but the downregulation of cyclins D1 and D3 by INK128 was more pronounced in *Snail* KO HCT116 cells (Fig. 6g), and these downregulations were not associated with changes in their transcript levels (Fig. 6h). Notably, the suppression of cyclins D1 and D3 was associated with increased levels of total and dephosphorylated 4E-BP1 in Snail-depleted cells, and their suppression was almost completely eliminated by silencing 4E-BP1 (Fig. 6g). Taken together, these results support a model whereby Snail-mediated repression of 4E-BP1 mitigates the anti-proliferative effect of mTORkis by continuous translation of cell cycle promoters such as cyclins D1 and D3.

**Targeting Snail enhances mTORki therapy in vivo.** To explore whether Snail-mediated 4E-BP1 repression also reduces the therapeutic response to mTORkis in vivo, nude mice bearing established *Snail* WT or *Snail* KO HCT116 xenograft tumors were treated with INK128 or vehicle control. Preliminary data showed that treatment with 1.5 mg kg$^{-1}$ INK128 effectively inhibited phosphorylation of the mTORC1 substrate 4E-BP1 and

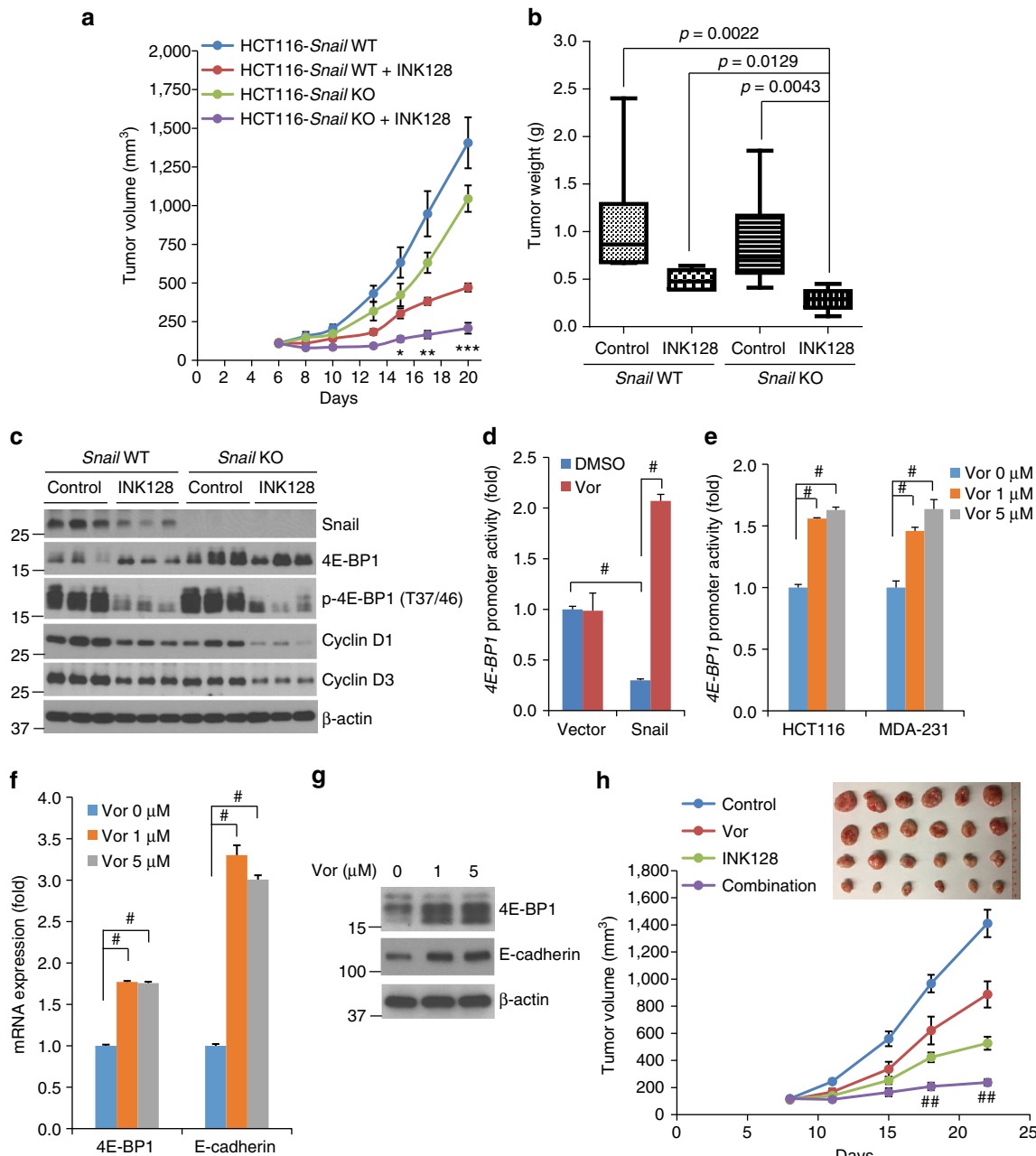

**Fig. 7** Co-targeting Snail and mTOR profoundly suppresses tumor growth in vivo. **a** Mice bearing HCT116-*Snail* WT or KO xenograft tumors were treated with INK128 at 1.5 mg kg$^{-1}$ or vehicle control twice per day for 5 consecutive days each week. The results are presented as the mean tumor volume ± SEM ($n = 6$ mice per group). *$P < 0.02$; **$P < 0.01$; ***$P < 0.001$ for INK128-treated *Snail* KO vs. INK128-treated *Snail* WT or vehicle control-treated *Snail* WT or KO using Student's *t*-test. **b** At the end of the experiment outlined in **a**, tumors were removed and weighed. The results are presented as the mean tumor weight ± SEM ($n = 6$ mice/group). Statistical significance was determine by Mann–Whitney test with the *p* value indicated. **c** Representative tumors from mice treated as in **a** were lysed 6 h after the final treatment with INK128 or vehicle control. Tumor lysates were analyzed by western blotting for the indicated proteins. **d, e** *4E-BP1* promoter activity was analyzed in T47D cells with stable expression of Snail or vector control **d**, and in HCT116 and MDA-231 cells **e** that were treated with vorinostat (Vor) for 12 h. **f, g** HCT116 cells were treated with vorinostat for 12 h **f** or 24 h **g**, followed by quantitative RT-PCR analysis for mRNA expression of 4E-BP1 and E-cadherin relative to the levels found in DMSO-treated control cells **f**, or by western blot analysis for the indicated proteins **g**. All graphic data are presented as mean ± SEM ($n = 3$ technical replicates per condition). #$P < 0.001$ using Student's *t*-test. **h** Mice bearing HCT116 xenografts were treated with vorinostat (50 mg kg$^{-1}$), INK128 (1.5 mg kg$^{-1}$), the combination of both drugs, or vehicle control once daily for 5 consecutive days each week. The results are presented as the mean tumor volume ± SEM ($n = 6$ mice per group). ##$P < 0.001$ for combination of vorinostat and INK128 vs. vorinostat, INK128, or control using Student's *t*-test

the mTORC2 substrate AKT for up to 8 h; however, phosphorylation rebounded by 12 h after treatment (Supplementary Fig. 11a). To sustain inhibition of mTOR downstream targets, we treated mice with 1.5 mg kg$^{-1}$ INK128 twice a day. Mice treated

with this regimen for 5 consecutive days/week for 3 weeks demonstrated no gross toxicity or significant weight loss (Supplementary Fig. 11b). Chronic treatment with INK128 slowed tumor growth in mice with *Snail* WT HCT116 xenografts,

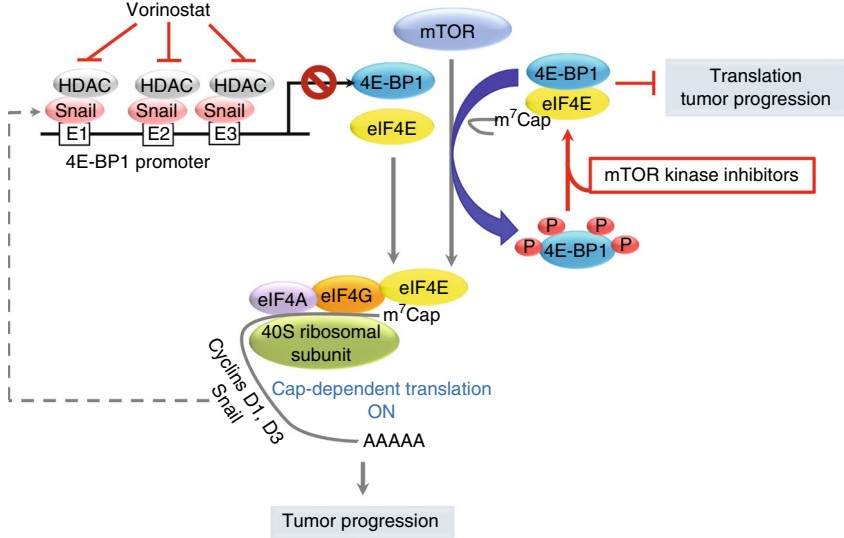

**Fig. 8** Snail modulates tumorigenesis and mTOR-targeted therapy by repression of 4E-BP1. Snail blocks *4E-BP1* gene transcription by binding to the sequences of three E-boxes within the *4E-BP1* promoter, which, in turn, promotes eIF4E-initiated cap-dependent translation of Snail itself and cyclins D1/D3 for cell proliferation, and reduces the therapeutic efficacy of mTOR kinase inhibitors. Genetic depletion of Snail or pharmacological inhibition of Snail co-repressor HDAC activity by vorinostat increases non-phosphorylated 4E-BP1 levels upon treatment with mTOR kinase inhibitors, which effectively enhances mTOR-targeted therapies via restoration of 4E-BP1 repressive function on cap-dependent translational control of tumor progression

whereas *Snail* KO alone provided a modest antitumor effect (Fig. 7a). However, INK128 dramatically suppressed growth in *Snail* KO-HCT116 xenografts (Fig. 7a, b). The marked growth suppression in vivo was associated with a pronounced increase in the level of the total and dephosphorylated 4E-BP1, as well as a dramatic decrease in the levels of cyclins D1 and D3 (Fig. 7c). These data recapitulate the increased resistance to mTORki by Snail expression seen in vitro.

It is known that Snail requires histone deacetylase 1 (HDAC1)/HDAC2 activities to repress the *E-cadherin* promoter[36]. To determine whether HDAC also functions as a co-repressor of Snail in repression of 4E-BP1, we used an Food and Drug Administration-approved pan-HDAC inhibitor, vorinostat[37], to test the potential effects of this inhibitor on modulation of 4E-BP1 expression and efficacy of mTORkis. First, we analyzed the effects of vorinostat on Snail-mediated repression of *4E-BP1* promoter activity. Similar to the data shown in Fig. 3b, ectopic expression of Snail in T47D cells reduced activity of the *4E-BP1* promoter to 30%, but this repression was totally eliminated by treatment with vorinostat (Fig. 7d). Furthermore, vorinostat treatment also significantly increased *4E-BP1* promoter activity in both HCT116 and MDA-231 cells, which express endogenous Snail (Fig. 7e). The increased promoter activity of *4E-BP1* was corroborated by increased *4E-BP1* transcripts and protein levels after vorinostat treatment in HCT116 cells (Fig. 7f, g). Consistent with the previous study[36], vorinostat treatment also markedly increased *E-cadherin* transcripts in HCT116 cells (Fig. 7f). These results strongly suggest that HDAC activity is also required for efficient suppression of the *4E-BP1* promoter. Next, we explored whether HDAC inhibition by vorinostat could mimic genetic depletion of Snail to improve the therapeutic effect of mTORkis. In the mouse model of HCT116 xenograft tumors with high Snail/low 4E-BP1 levels, treatment with either vorinostat (50 mg kg$^{-1}$) or INK128 (1.5 mg kg$^{-1}$) modestly decreased tumor size (Fig. 7h). However, vorinostat in combination with INK128 almost completely suppressed tumor growth with only marginal weight loss (e.g., < 10%) in mice (Fig. 7h and Supplementary Fig. 11c). These findings indicate that HDAC inhibitors greatly enhance the antitumor effect of mTORkis, suggesting that combined inhibition of HDAC and mTOR may be an effective

therapeutic strategy in cancers that overexpress Snail with decreased 4E-BP1 expression.

## Discussion

Snail is a key transcriptional factor with a well-characterized function in promoting EMT, cancer cell invasion, and metastasis[38,39]. Beyond EMT and tumor invasiveness, Snail and other EMT-inducing transcription factors cooperate with oncogenes in malignant transformation, regulate cancer cell stemness and differentiation, contribute to cancer cell survival and metabolic reprogramming, and impart resistance to chemotherapy[35,40–42]. In this study, we uncover a novel function of Snail in mTOR/4E-BP1-mediated translational control of tumorigenesis. We identified 4E-BP1 as a conserved transcriptional target of Snail in mammals. We show that Snail represses 4E-BP1 expression to promote the recruitment of capping mRNAs and ribosomal subunits to the eIF4F translation initiation complex for active translation in polysomes. Targeted inhibition of Snail activity increases 4E-BP1 levels, which is required to enhance the sensitization of cancer cells to mTOR kinase inhibitors. Our work suggests that expression of Snail with a concomitant reduction of 4E-BP1 accounts for the limited efficacy of mTOR inhibitors in cancer therapy (Fig. 8).

Our study provides several insights into the biology and therapeutic relevance of Snail in the translational control of tumor progression. First, our study reveals Snail as a positive regulator of cap-dependent translation via repression of 4E-BP1, suggesting that a Snail-specific translational program exists during malignant transformation, EMT, and tumor progression. We previously demonstrated that loss of 4E-BP1 induces EMT, increases breast and colorectal cancer cell motility and invasiveness, and promotes metastasis by upregulation of Snail expression in a cap-dependent translational manner[14]. Loss of 4E-BP1 can result in an increased level of free eIF4E and phosphorylated eIF4E to promote activation of cap-dependent translation[13,43]. Similar to 4E-BP1 loss, eIF4E phosphorylation also promotes EMT and metastasis by a translational increase of Snail protein[44]. Our present data showed that Snail acts as a reciprocal feedback repressor of 4E-BP1 to promote eIF4E-initiated capping mRNA

translation and protein synthesis. Together, these results suggest the existence of a positive feedback loop for continuous translation of Snail through repression of 4E-BP1 (Fig. 8). This positive feedback loop might be required to maintain Snail expression to transcriptionally but also translationally regulate expression of a variety of gene sets responsible for the malignant transformation, EMT induction, and metastasis. Whether Snail transcriptionally and translationally regulates genes through a differential or cooperative manner to alter cell plasticity for tumor progression is likely to be complex and a matter for further investigation. Moreover, our study found that the Snail family member, Slug, also represses *4E-BP1* promoter activity and its expression, suggesting that Slug may have secondary effect on cap-dependent translation. Whether tumors with co-expression of Snail and Slug require both proteins to additively initiate a cap-dependent translation program during EMT and cancer development is another question that remains to be determined.

Second, our study reveals Snail as an important determinant of cancer cell sensitivity to mTOR inhibitors. Deregulation of cap-dependent translation by mTOR-mediated phosphorylation of 4E-BP1 plays an important role in cancer progression[17,29,30,45]. Thus, the use of mTORkis to inhibit 4E-BP1 phosphorylation presents a promising strategy for cancer therapy. However, loss of 4E-BP1 expression with an increase in free eIF4E, rather than the phosphorylation status of 4E-BP1, is considered to be a primary determinant for cancer cell resistance to mTORkis[12,13]. Here we identified Snail as a strong transcriptional repressor of 4E-BP1, comparable to the effect of this repressor on E-cadherin expression. Our study demonstrates that the Snail-regulated 4E-BP1 level is a critical determinant of the anti-proliferative effects of mTORkis through cap-dependent translational control of cyclins D1 and D3 expression, and independent of AKT/S6K activation. Our findings are consistent with several studies showing a positive role for Snail in cell proliferation by using a Snail transgenic mouse skin model, expression or disruption of Snail in the *Drosophila* ovarian model, and the PTEN-deficient glioblastoma cells with knockdown of Snail expression[46–49]. However, others reported opposite results that ectopic expression of Snail in the canine normal epithelial MDCK cell line using a single cell clone undergoing EMT could induce G1 arrest and resistance to apoptosis by transcriptional repression of cyclin D2 and PTEN expression[21,34]. Distinct from these reports[21,34], we utilized a population pool of the stable Snail-expressing cells in which Snail expression may not be high enough to induce a complete EMT. Indeed, using individual clones with different expression levels of Snail in MDCK cells, we were able to reproduce their findings[21,34] and found that high levels of Snail expression is required to maximally repress expression of E-cadherin, cyclin D2, and PTEN; induce a complete EMT; inhibit cell cycle progression and proliferation; and render cell resistance to serum deprivation-induced apoptosis (Supplementary Fig. 12a–g). Nevertheless, it is noteworthy that 4E-BP1 expression in MDCK cells could be effectively inhibited even at low to medium Snail expression levels. In addition, these cells showed a partial EMT with no or weak inhibition of expression of E-cadherin, cyclin D2, and PTEN, but significantly increased cell proliferation and G1/S progression, and were resistant to apoptosis (Supplementary Fig. 12a–g). Similar results were observed in the mixed stable Snail-expressing MDCK cell populations that showed a moderate Snail expression level and relieved the anti-proliferative effect of mTORkis (Supplementary Fig. 12h–j). Thus, the discordant Snail-effect on cell proliferation and survival is probably associated with diverse effects in the cell population expressing different levels of Snail with variable EMT status and the type of cell or tissue and species used. Another possibility is that in cancer cells, Snail cooperates with other oncogenic signaling molecules, such as

mTOR, to maximally downregulate and inhibit their common target, 4E-BP1, as shown here, to promote cell proliferation and tumor progression by translational regulation of cyclins D1 and D3 expression, and other potential targets. Together, these processes allow cancer cells to escape mTOR-targeted therapies.

Third, our study strongly suggests that inhibition of Snail activity could largely improve therapeutic efficacy of mTORkis in cancers with low 4E-BP1 levels. Overexpression of Snail is frequently observed in advanced carcinomas including breast and colorectal cancers[39,50,51]. Our discovery of Snail as an important modulator of 4E-BP1 expression and cancer cell sensitivity to mTORkis suggests that the limited preclinical and clinical efficacies of mTOR inhibition in breast, colorectal and other cancers[52] may be attributed, in part, to the frequent genetic gains by Snail associated with decreased expression of 4E-BP1. Our findings indicate that Snail expression may serve as a predictive marker for mTOR-targeted therapies in cancer. Extensive evidence indicates that Snail is an attractive therapeutic target; yet, there are no small molecules to inhibit Snail's functions. As Snail is required to recruit multiple chromatin enzymes for its transcriptional repressive function[53,54], targeting these epigenetic regulators such as HDAC could be an alternative approach to inhibit Snail activity. In this study, we found that, similar to genetic depletion of Snail, pharmacological inhibition of HDAC activity significantly repressed Snail activity by inducing 4E-BP1 expression, and co-targeting HDAC and mTOR resulted in a marked suppression of tumor growth. Several recent studies[55–57] also show that the combination of mTOR and HDAC inhibitors exert a synergistic antitumor activity in a large panel of human cancer cell lines, and the patient-derived xenograft and transgenic mouse models. Interestingly, the synergistic growth inhibiting consequence of combined mTOR/HDAC inhibition is likely attributed to their mechanistic convergences on the mTOR/4E-BP1 signaling axis and impaired polysome formation[56]. This situation is similar to our findings obtained with Snail depletion in combination with mTORkis. Given that multiple clinical trials involving mTOR inhibitors as monotherapy or in combination with HDAC inhibitors are being tested in advanced cancers (NCT01058707, NCT02719691, NCT01087554, and NCT00918333), our findings suggest that incorporating the analysis of Snail and 4E-BP1 expression in primary tumors may help to prospectively identify resistance to mTOR inhibitors in the clinic studies.

## Methods
**Cell culture**. Human breast (H1500, MCF7, T47D, ZR75-1, MDA-231, and MDA-157), colon (HCT116, DLD-1, HCT15, and SW480), and lung (A549) cancer cell lines, mouse NMuMG, 4T1 and NIH3T3 cell lines, and the MDCK cell line were obtained from the American Type Culture Collection (ATCC, Manassas, VA) and cultured in the appropriate medium with supplements as recommended by ATCC. The SUM149 human breast cancer cell line was obtained from Asterand (Detroit, MI) and cultured in Ham's F-12 supplemented with 5% fetal bovine serum (FBS), 5 μg ml$^{-1}$ insulin, and 1 μg ml$^{-1}$ hydrocortisone. The IMR-90 and TIG-1 normal human fetal lung fibroblasts were obtained from Coriell Institute for Medical Research (Camden, NJ) and cultured in Eagle's minimum essential medium supplemented with 15% FBS. All cell lines were tested for mycoplasma contamination via PCR (e-Myco Plus kit; iNtRON Biotechnology, Kirkland, WA) and were found to be negative. In addition, all cell lines are routinely checked for morphologic and growth changes, to probe for cross-contamination or genetically drift. If present, cell lines were re-authenticated using the short tandem repeat profiling service by ATCC.

**Plasmids and reagents**. The pLenti6.3-human Snail and Twist1 plasmids were generated as described previously[35,58]. Human Slug and Zeb1 were also cloned into the same pLenti6.3 lentivirus expression vector (Invitrogen, Grand Island, NY). To establish stable transfectants with specific protein expression, cells were infected with lentivirus using the indicated pLenti6.3 constructs followed by selection with puromycin (2 μg ml$^{-1}$) for 7–10 days as described previously[30]. The human *4E-BP1* promoter (positions − 1,555/+233) and mouse *4E-BP1* promoter (positions − 1,430/+80) were amplified from HCT116 and NIH3T3 cell genomic DNAs,

respectively, and then cloned into the pGL3 basic reporter vector that contains firefly luciferase (Promega, Madison, WI). The mutant reporter constructs that carry the *4E-BP1* promoter with combinations of the three mutated E-boxes (see Fig. 3a) were generated using the QuikChange XLII site-directed mutagenesis kit (Stratagene, La Jolla, CA). The primers used are listed in Supplementary Table 1. All sequences were verified by automated DNA sequencing. The pBabe-4E-BP1 plasmid with hygromycin resistance was generated as described previously[14,29]. The pcDNA3-HA-AKT1 plasmid (73408) and p3xFLAG-mSnail plasmid (34583) were purchased from Addgene (Cambridge, MA). AZD8055 and INK128 were obtained from Selleckchem (Houston, TX) and Active Biochem (Maplewood, NJ), respectively. Vorinostat was purchased from MedChem Express (Monmouth Junction, NJ).

**Gene silencing by siRNA and shRNA.** Smart pool siRNA against human Snail (L-010847), Slug (L-017386), Twist1 (L-006434), Zeb1 (L-006564), 4E-BP1 (L-003005), or non-targeting control siRNA pool (D-001810-10) was obtained from Dharmacon (Chicago, IL). Cells were transfected with 20 nM siRNA pool against the indicated genes or control siRNA pool using Lipofectamine RNAiMAX reagent according to the manufacturer's instructions (Invitrogen). After 36–48 h transfection, cells were subjected to assays as indicated. The lentiviral shRNAs against human and mouse Snail were cloned into pLKO.1 vector (Sigma, St Louis, MO), and their sequences are listed in Supplementary Table 2. The Non-Target Control shRNA (SHC002) and the lentiviral shRNA against human 4E-BP1 were from Sigma. The specificity of the 4E-BP1 targeting sequence has been verified in our previous study[30]. The lentiviral shRNAs against human Slug were obtained from Addgene (10903; 10904). To establish stable transfectants with knockdown of specific protein expression, cells were infected with lentivirus using the indicated shRNA constructs followed by selection with puromycin (2 µg ml⁻¹, for Snail and Slug shRNAs) or hygromycin (250 µg ml⁻¹, for 4E-BP1 shRNA) for 7–10 days[30].

**Generation of human *Snail* KO cell lines.** Two small guide RNAs (sgRNAs) were designed to target exon 1 of human *Snail* at two different sites using the CRISPR designing tool[59] and their sequences are listed in Supplementary Table 2. The distance between target sites is roughly 50 bp. The two sgRNAs were cloned into the Cas9 D10A nickase-coding pX462 vector (Addgene), and transiently co-transfected into HCT116 and MDA-231 cells using Lipofectamine 3000 according to the manufacturer's protocol (Invitrogen). For *Snail* WT cells, HCT116 and MDA-231 cells were transfected with non-sgRNA-Cas9 D10A nickase-encoding pX462 vector. Single cells were selected by serial dilution followed by puromycin (2 µg ml⁻¹) treatment for 1 week. Single-cell colonies were screened for internal deletion by sequencing the PCR fragments using the following primers: 5′-CCCAGTGATGTGCGTTTCCC-3′ (forward) and 5′- CCCAACCACCCAGACA-GATC-3′ (reverse).

**Quantitative real-time PCR analysis.** Total cellular RNA was isolated using the RNeasy plus mini kit (Qiagen, Valencia, CA). Equal amounts of RNA were used as templates for all reactions. Complementary DNA was generated with the Super-Script III First Strand Synthesis System (Invitrogen). Real-time PCR was performed on a StepOne Real-Time PCR system (Applied Biosystems, Foster City, CA) in triplicate with Maxima SYBR Green/ROX qPCR Master Mix (ThermoFisher Scientific, Waltham, MA). The PCR primers are listed in Supplementary Table 3. Glyceraldehyde 3-phosphate dehydrogenase was used as an internal control for normalization and relative expression level was calculated by the comparative CT ($^{\Delta\Delta}$CT) method. Each experiment was performed in triplicate and repeated at least three times.

**4E-BP1 promoter reporter assay.** Cells ($1 \times 10^5$) were co-transfected with 0.5 µg of the *4E-BP1* promoter or its mutant construct together with 0.1 µg of pHRL-TK *Renilla* luciferase control vector using Lipofectamine 3000 according to the manufacturer's protocol (Invitrogen). Thirty-six hours post transfection, firefly and *Renilla* luciferase activities were measured using a dual-luciferase assay kit (Promega). The firefly luciferase activity for each sample was normalized based on transfection efficiency as determined by *Renilla* luciferase activity. Each experiment was performed in triplicate and repeated at least three times.

**ChIP assay.** ChIP assay was performed according to the protocol described by Nowak et al.[60] with some modifications. Cells ($4–6 \times 10^6$) were cross-linked with 1% formaldehyde for 15 min at room temperature, then lysed in L1 buffer (50 mM Tris-HCl, pH 8.0, 2 mM EDTA, 0.1% IGEPAL, 10% glycerol, 1 mM dithiothreitol (DTT), 1 mM phenylmethylsulfonyl fluoride, and protease inhibitor cocktail) on ice. Nuclei were pelleted by centrifugation and resuspended in ChIP lysis buffer (50 mM Tris-HCl, pH 8.0, 10 mM EDTA, 1% SDS). Chromatin was subjected to sonication and then immunoprecipitated with 2 µg of Snail antibody (AF3639, R&D Systems, Minneapolis, MN) or an irrelevant immunoglobulin G (IgG) overnight, followed by incubation with a 50% slurry of protein G sepharose/salmon sperm DNA (Invitrogen) for 3 h at 4 °C. Bound DNA–protein complexes were eluted, and crosslinks were reversed after a series of washes. Purified DNA was resuspended in TE buffer (10 mM Tris–HCl, pH 8.0, 1 mM EDTA) for PCR

analysis. The primers used for the *4E-BP1* and *E-cadherin* promoters are listed in Supplementary Table 4.

**Cap-dependent translation assay.** Cells ($8 \times 10^4$) were transfected with a bicistronic luciferase reporter plasmid (0.2 µg), pcDNA3-rLuc-PolioIRES-fLuc, which directs cap-dependent translation of the *Renilla* luciferase gene and cap-independent Polio IRES-mediated translation of the firefly luciferase gene[29]. After 24 h transfection, cells were treated with AZD8055 or INK128 for 12 h and cell lysates were assayed for *Renilla* and firefly luciferase activities using a dual-luciferase assay kit (Promega). Cap-dependent *Renilla* luciferase activity was normalized against cap-independent firefly luciferase activity as the internal control. The ratio of *Renilla*/firefly luciferase activity was calculated for cap-dependent translational activity[28,29]. Each experiment was performed in triplicate and repeated at least three times.

**Immunoprecipitation and western blot analysis.** Cells were lysed in NP-40 lysis buffer (50 mM Tris-HCl, pH 7.5, 150 mM NaCl, 1 mM EDTA, 1% NP-40, 10% glycerol, protease, and phosphatase inhibitor cocktail). The cell lysates (250 µg protein) were immunoprecipitated with 1 µg of eIF4E antibody (sc-271480, Santa Cruz Biotechnology, Dallas, TX) overnight followed by incubation with a 50% slurry of protein G sepharose beads for 3 h at 4 °C. The beads were washed three times with the lysis buffer and the immunoprecipitated protein complexes were resuspended in 2× Laemmli sample buffer followed by western blot analysis. Western blot analysis were performed using equivalent total protein loadings as described previously[29]. Antibodies for p-Akt (Ser473) (#4060; 1: 1000), p-p70S6 Kinase (Thr389) (#9234; 1: 1000), p-S6 (Ser235/236) (#4858; 1: 1000), p-4E-BP1 (Thr37/46) (#2855; 1: 1000), p-4E-BP1 (Ser65) (#13443; 1: 1000), p-4E-BP1 (Thr70) (#13396; 1: 1000), 4E-BP1 (#9644; 1: 1000), eIF4E (#2067; 1: 1000), Snail (#3879; 1: 1000), Slug (#9585; 1: 1000), and Zeb1 (#3396; 1: 1000) were from Cell Signaling Technology (Danvers, MA). Cyclins D1 (sc-718; 1: 1000), D2 (sc-181; 1: 500) and D3 (sc-182; 1: 1000), Twist (sc-81417; 1: 50), PTEN (sc-7974; 1: 500), eIF4G (sc-133155; 1: 1000), and eIF4A (sc-377315; 1: 1000) antibodies were from Santa Cruz Biotechnology. E-cadherin antibody (#610181; 1: 20,000) was from BD Biosciences (San Jose, CA) and β-actin antibody (A5411; 1: 10,000) was from Sigma. Uncropped scans of the most important western blottings are shown in Supplementary Fig. 13.

**Cap-binding assay.** Cap-binding assay was performed as described previously[29]. Briefly, cell lysates (500 µg protein) as prepared in the NP-40 lysis buffer were incubated at 4 °C overnight with m⁷GTP Sepharose beads (GE Healthcare Life Sciences, Pittsburgh, PA) to capture eIF4E and its binding partners. Precipitates were washed three times with the lysis buffer and resuspended in 2× Laemmli sample buffer followed by western blot analysis.

**Polysome analysis.** Polysome analysis was performed as described previously[29] with some modifications. Sucrose density gradient centrifugation was employed to separate the ribosome fractions following treatment of cells with drugs. 15 min before collection, cycloheximide (100 µg ml⁻¹) was added to the culture medium. Cells were washed with ice-cold phosphate-buffered saline (PBS) containing 100 µg ml⁻¹ cycloheximide and collected in polysome lysis buffer (5 mM Tris-HCl, pH 7.5, 2.5 mM MgCl₂, 1.5 mM KCl, 2 mM DTT, 0.5% Triton X-100, 0.5% sodium deoxycholate, 100 µg ml⁻¹ cycloheximide, 200 U ml⁻¹ RNAsin, 0.2 mg ml⁻¹ heparin, and protease inhibitors). Cells were incubated on ice for 15 min and then centrifuged at $10,000 \times g$ for 10 min at 4 °C. The supernatant (4 mg protein) was layered on a pre-chilled 10–50% linear sucrose gradient preparing in the gradient buffer (5 mM Tris-HCl, pH 7.5, 2.5 mM MgCl₂, 1.5 mM KCl, 2 mM DTT, 100 µg ml⁻¹ cycloheximide, 40 U ml⁻¹ RNAsin, 0.1 mg ml⁻¹ heparin, and protease inhibitors), and then centrifuged in a Beckman SW40Ti rotor at $250,000 \times g$ for 2.5 h at 4 °C. Gradients were fractionated, while monitoring absorbance at A254 with a Gradient Station System (Biocomp, Fredericton, NB, Canada).

**Protein synthesis assay.** To determine nascent protein synthesis, Click-iT AHA (L-azidohomoalaine), Alexa Fluor 488 alkyne, and Click-iT cell reaction buffer kit were purchased from Invitrogen and used according to the manufacturer's protocol. Briefly, Click-iT AHA (50 µM), a methionine analog containing an azide moiety, was added to cells in methionine-free medium (Invitrogen) for 1 h. The cells were washed twice with PBS, fixed with 4% paraformaldehyde, and permeabilize with 0.25% Triton X-100. Detection of the incorporated amino acid utilizes a click reaction between an azide and alkyne, where the azido-modified protein is detected with an Alexa Fluor® 488 by flow cytometry.

**Cell growth and proliferation assays.** Cell growth was assessed as described previously[30]. Briefly, $5 \times 10^4$ cells were seeded in six-well plates in triplicate. After 24 h, cells were treated with the indicated drugs and incubated at 37 °C. The cells were cultured for 3 days and the number of viable cells was counted using the Vi-CELL XR 2.03 (Beckman Coulter, Brea, CA). Cell proliferation was determined by incorporation of EdU (5-ethynyl-2′-deoxyuridine) during DNA synthesis using a

Click-iT EdU imaging kit according to the manufacturer's protocol (Invitrogen). Each experiment was performed in triplicate and repeated at least three times.

**Colony formation assay.** Cells were seeded in 12-well plates at 500 cells per well, followed by the treatment with the indicated drugs the next day. Fresh culture medium containing the corresponding concentrations of the drugs or dimethyl sulfoxide (DMSO) vehicle control was added every 3 days. After 12 days of treatment, the medium was removed, and cell colonies were fixed with 4% paraformaldehyde and stained with 0.1% crystal violet.

**Cell cycle and apoptosis analysis.** Cells were plated in 100 mm dishes. After 24 h, cells were treated as indicated in figure legends; both adherent and floating cells were collected. For cell cycle analysis, cell nuclei were prepared by the method of Nusse[61] and stained with ethidium bromide[62]. Cell cycle phase distribution was determined by flow cytometry. For apoptosis, cells were analyzed by flow cytometry using the Annexin V-APC Apoptosis Detection Kit according to the manufacturer's protocol (ThermoFisher Scientific).

**Immunohistochemical staining.** Paraffin-embedded colorectal cancer tissue sections were obtained from the tissue bank of Nanfang Hospital at Southern Medical University in China. The tissue sections were deparaffinized, rehydrated and treated with hydrogen peroxide. Antigen retrieval was performed using citrate buffer (pH 6.0) in a steamer. Tissue sections were first blocked with avidin, biotin, and 5% normal goat serum and then incubated in a humidified chamber at 4 °C overnight with antibodies against Snail (NBP1-19529; 1: 100; Novus Biologicals, Littleton, CO) or 4E-BP1 (#9644; 1: 2,000; Cell Signaling Technology). The samples were then incubated with biotin-labeled goat anti-rabbit secondary antibody and subsequently with horseradish peroxidase–avidin complex (Vector Laboratories, Burlingame, CA). Antibody-associated staining was visualized using diaminobenzidine substrate solution and the tissue sections were then counterstained with hematoxylin. The immunoreactivity was scored blindly according to the value of immunoreaction intensity and the percentage of tumor cell staining using a semiquantitative seven-tier system[63,64].

**Animal studies.** Male athymic nude mice (5–6 weeks old) were purchased from Taconic (Hudson, NY), and maintained and treated under specific pathogen-free conditions. Experiments were carried out under a protocol approved by the University of Kentucky Institutional Animal Care and Use Committee. HCT116 xenograft tumors were established by subcutaneously injecting HCT116 cells ($3 \times 10^6$ per mouse) in a 1:1 mixture of media and Matrigel (BD Biosciences). Mice were randomized among control and treated groups ($n = 6$ per group) when tumors were well-established (~150–180 mm³). INK128 was formulated in 5% $N$-methyl-2-pyrrolidone plus 15% polyvinylpyrrolidone as described[10] and delivered orally at 1.5 mg kg⁻¹ once or twice per day for 5 consecutive days each week. Vorinostat was dissolved in DMSO and diluted in 1:1 PEG400 and sterile water to a final composition of 10% DMSO, 45% PEG400, and 45% water as described[65]. Vorinostat was administered intraperitoneally at 50 mg kg⁻¹ once per day for 5 consecutive days each week. For combination treatment, both drugs were given concurrently. Control mice received vehicle alone for both drugs. Tumor dimensions were measured using a caliper and tumor volumes were calculated as mm³ = π/6 x larger diameter x (smaller diameter)². Tumors were excised and snap frozen in liquid nitrogen, homogenized in 2% SDS lysis buffer, and then processed for western blot analysis[29,30].

**Statistical analysis.** Statistical analyses for each experiment were performed as described in the corresponding figure legends. Data between groups were compared using a two-tailed unpaired Student's $t$-test, $\chi^2$-test, or Mann–Whitney test. All data are presented as mean±SEM, if not stated otherwise. Differences between groups were considered statistically significant at $P < 0.05$. Data presented is representative of two or more independent experiments, unless indicated otherwise.

**Data availability.** All data generated or analyzed during this study are available within the Article and Supplementary Files, or available from the corresponding author upon reasonable request.

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

## Acknowledgements

We thank the Markey Cancer Center Research Communication Office for assistance with editing this manuscript. This work was supported by grants from NIH R01CA175105 and NIH R01CA203257 (Q.-B.S.). We also acknowledge use of the UK Flow Cytometry & Cell Sorting core facility, supported in part by the Office of the Vice President for Research and the Markey Cancer Center NCI Center Core Support Grant (P30CA177558).

## Author contributions

J.W., Q.Y., and Y.C. designed and performed most of the experiments and analysed data. Y.G. and S.L. performed the immunohistochemical staining analysis. X.H. and W.M. performed part of CRIPR and shRNA experiments. C.W. contributed to the statistical analysis. H.-S.Y. contributed to part of polysome analysis. B.P.Z. and B.M.E. provided critical reagents and comments, and discussed the results. Q.-B.S. conceived and designed the study, analysed and interpreted data, and wrote the manuscript.

## Additional information

**Competing interests:** The authors declare no competing financial interests.

