## [Peer Review File · Nature Communications]

Reviewers' comments:

Reviewer #1 (Remarks to the Author):

In this article the authors describe the regulation by Snail1 of 4E-BP1, cap-dependent translation inhibitor. The repressive activity of 4E-BP1 is abrogated by its phosphorylation by TORC1 kinase. The authors adequately demonstrate that Snail1 represses 4E-BP1 transcription and down-regulates its expression, inducing resistance to mTORC kinase inhibitors. These experiments are adequately designed and performed and the conclusions of the first part (regulation of 4E-BP1 by Snail1) are solid. However, since Snail1 also activates other proteins regulated by TORC such as Akt (see Vega et al, 2004 or Escriva et al 2008, op. cit) it is not clear to me that the increased resistance to TORC inhibitors detected in Snail1-expressing cells is due uniquely (or mainly) to the repression of 4E-BP-1 and not to the activation of Akt. The authors should perform additional experiments to study the different contribution of these two elements: 1) they should analyze Akt activity in cells treated with TORCKis, either expressing Snail1 or not; 2) they should determine TORCKis sensitivity in Snail1-expressing cells transfected with 4E-BP1; and 3) in Snail1-depleted cells they should down-regulate 4E-BP1 or over-express Akt and analyze the different contribution of both elements to TORCKis sensitivity. These experiments will greatly increase the general interest of the article.

Other

1) It is accepted in the field that Snail1 overexpression in epithelial cells decreases cell proliferation as initially reported by Vega et al 2004, who indicated that Snail1 repressed cyclinD2 promoter. However, for the authors, Snail1 expression increases proliferation in MCF7 and T47D cell lines and do not alter cyclin D2, as mentioned in the Discussion. They have tried to explain this discrepancy indicating that it might be matter of cells specificity. I agree with them if we compare mesenchymal with epithelial cells but not when expressing Snail1 in epithelial cells; since 2004 many other groups have reproduced Vega's result in other cell lines. Is the effect observed by the authors also reproduced in MDCK cells? Is this a consequence of lower expression of Snail1 compared with that in Vega et al experiments? Does their Snail1 construct hold any mutation that modifies its capability to repress? The authors should try to perform some assays to solve this conflict, not merely indicating that it is a matter of cell line because the same argument might also apply to their results.

2) The authors should include a more detailed characterization of the commercial 4E-BP1 antibody (Novus) they have used for the IHC analysis since it is different from the one used in WB that seems to be specific.

3) Since they indicate that eIF4E/4E-BP1 binding is modified by Snail1, they should show this by a CoIP assay; a m7GTP pull down does not substitute this analysis.

4) Other factors have been shown to mimic or extend Snail1 repression of E-Cadherin. It would be useful for the researchers working in this area to know if Slug, Zeb1/2 or Twist also modulate 4E-BP1 levels and repress this promoter.

5) Snail1 is generally accepted as a short range transcriptional repressor. In 4E-BP1 promoter some of the E-boxes, putative binding sites for Snail1, are more than 1000 bp upstream from the transcription start site, what is surprising. The ChIPs presented for Snail1 binding to the three boxes in Figure 3d should be quantified and the results presented as "percentage of input" to provide a better determination of the relative binding.

6) The inhibitor of 4E-BP1 phosphorylation INK128 substantially decreases Snail1 levels (see Figure 7C) likely due to the inhibitory effect of 4E-BP1 on Snail1 expression. However, this down-regulation in Snail1 is not accompanied by a 4E-BP1 increase. This discrepancy on the model

should be responded. It is possible that the levels of Snail1 bound to 4E-BP1 Promoter are not modified by INK128 since this pool of Snail1 is probably more stable than the DNA-unbound Snail1. This hypothesis should be determined by a ChIP experiment of Snail1 in the presence or absence of the inhibitor.

7) The authors have reported that Snail1 expression is dependent on 4E-BP1 and 4E-BP1 phosphorylation (Cai et al, *Oncotarget* 2014, ref 17). This inhibition should be included in the model depicted in Figure 8 and the two reciprocal effects of Snail1 and 4E-BP1 better discussed in the appropriate section.

Reviewer #2 (Remarks to the Author):

In this manuscript entitled 'Snail determines the therapeutic response to mTOR kinase inhibitors by transcriptional repression of 4E-BP1' the authors identify the zinc finger transcription factor Snail as a novel transcriptional regulator of 4EBP1. 4EBP1 an important negative regulator of cap-dependent protein synthesis via its interaction with the cap binding protein, eukaryotic translation initiation factor 4E (eIF4E). Several key oncogenes and tumor suppressors converge on eIF4E or its regulatory partners in order to modulate cap-dependent protein synthesis in cancer cells. Therefore, therapeutic agents that target components of the protein synthesis apparatus hold tremendous promise as anti-cancer drugs and a large emphasis has been placed on the development of therapeutics or novel treatment strategies to target the protein synthesis machinery in cancer including the use of mTOR inhibitors.

Importantly, Wang and colleagues present findings to suggest that Snail levels may regulate the therapeutic response to mTOR inhibitors by repressing 4EBP1 using a number of different cancer cell lines and in vitro cell proliferation assays. The authors clearly demonstrate that Snail selectively downregulates 4EBP1 expression at the transcription level and not other 4EBP family members by binding to specific sequences in the promoter of 4EBP1. While these findings may be novel, it is not clear whether this mechanism of regulation of 4EBP1 by Snail is evolutionarily conserved and is not an artifact of the cell models used. The clinical significance of Wang et al., findings is that Snail expression and function may modulate the response to mTOR inhibitors that are currently used in clinical trials. However, the authors often overstate the effect of modulating Snail on the efficacy of mTOR inhibitors in the cell lines tested and in xenografts, which is at best modest, and fail to provide mechanistic insights into how 4EBP1 regulates cancer cell sensitivity to mTOR inhibitors in an unbiased manner. Importantly, it has been previously published that 4EBP1 levels modulate sensitivity to mTOR inhibitors. Furthermore, the clinical significance of this study is not well discussed or experimentally investigated. Therefore, in the current form these findings do not advance the current understanding of the importance or relevance of Snail transcriptional effects on 4EBP1 expression and activity in regards to cancer treatment and this reviewer believes the manuscript is not suitable for publication in *Nature Communications*. Below are some suggestions that the authors may find useful for submission to a more specialized journal.

Major points:

1. Snail is an evolutionarily conserved zinc finger transcription factor involved in mesoderm formation in vertebrates. In cancer, Snail has a well-documented role in binding sequence specific E box region of promoters to repressor transcription of factors modulating epithelial-mesenchymal transition (EMT) and cell survival. In this manuscript, Wang et al., identify 4EBP1 as a previously uncharacterized transcriptional target of Snail. While the findings demonstrating that Snail directly influences the transcription of 4EBP1 are convincing, there is a strong concern that these affects

may be an artifact of the cell models used. The authors should therefore confirm their findings that 4EBP1 is a conserved transcriptional target of Snail in mouse and also in a more physiological system such as normal human fibroblasts.

2. Wang and colleagues present a number of figures (4-7) supporting a role of Snail modulation of 4EBP1 levels which in turn regulate cap-dependent translation and the therapeutic response of cancer cells to mTOR inhibitors. Several panels relating to figures 4-6 are redundant and would normally be considered supplementary data. Furthermore, at times, the authors overstate their findings on the effect of Snail expression on the efficacy of mTOR inhibitors, which is quite modest. For example, the authors' statement that 'Repression of 4E-BP1 expression by Snail renders cancer cell resistant to mTORkis' is misleading, as in Figure 5a-b the authors are comparing distinct cancer cell lines that harbor numerous genetic alterations and are from different tissue of origin. Moreover, in Figure 5C cells expressing exogenous Snail are sensitivity to mTOR inhibitors when compared to isogenic control cells. Therefore, it is recommended that Wang et al., revise these points and clearly modify the text so as to accurately reflect the data presented.

3. In Figure 4, the authors demonstrate that 'Snail-mediated repression of 4E-BP1 promotes cap-dependent translation and relieves the inhibitory effect of mTORkis on translation'. The authors further demonstrate that 'Snail mitigates the anti-proliferative effect of mTORkis by translational control of cell cycle regulators such as cyclins D1 and D3'. However, Wang and colleagues have not taken into consideration that changes in gene expression at the translation level that are cap-independent may also modulate the sensitivity to mTOR inhibitors upon Snail repression and that targets other than Cyclin D1 and D3 may contribute to the observed phenotype. Therefore, the authors should undertake different approaches such as using reporter constructs to monitor changes in cap-independent translation.

4. Relating to the last point, it would be very important to determine whether a Snail-specific translational program exists during physiological conditions where Snail is known to play an important role, for example during EMT.

5. While the authors' findings indicate that Snail expression may serve as a predictive marker for mTOR therapies, the authors do not provide experimental evidence of the clinically significant of their findings using a more physiologically relevant system. For example, the authors should test whether inhibition of co-repressors of Snails transcriptional activity such as HDAC1 and HDAC2 may serve as efficient therapy in cancers with low 4EBP1 activity and in combination with mTOR inhibitors. Such experiments would further support the use of Snail as a putative modulator of 4EBP1 and mTOR inhibitors in cancer therapy.

Minor points:

1. On page 7 line 167, the sentence 'Conversely, knockdown of Snail expression in HCT116 and MDA-231 cells repressed the translation activity, which was enhanced the inhibitory effect of AZD8055 or INK128 on translation (Fig. 4b).' should be revised for clarity.

2. In Figure 5, panel b the word 'Sanil' should be corrected to 'Snail'.

Reviewer #3 (Remarks to the Author):

The manuscript by the She group uncovers a regulation of 4EBP1 expression by the Snail

transcription factor. In a number of cell lines and overexpression/knock-down studies, they convincingly show that Snail expression inversely correlates with 4EBP1 expression. They claim that this is due to a direct control of Snail on the 4EBP1 promoter and that this determines the sensitivity of cancer cells to treatments with mTOR inhibitors. These two points requires further experimental evidence. Otherwise the study is interesting and may be relevant for growth control and tumorigenesis.

1) It is unclear how fast and direct the effect of Snail on 4EBP1 transcription is. I could not easily find the information how long one has to wait to see an effect of Snail modulation on 4EBP1 expression.

2) The major drawback of the study is that the authors did not measure mTOR activity after Snail overexpression and knockdown. In figure 6A and 6B, it looks like Snail KO and rescue have major effects on 4EBP1 band shift. This might indicate increased mTOR activity upon Snail gain-of-function. This should be directly tested by measuring the phosphorylation of other mTOR substrates, including S6K and Akt. It is possible that the sensitivity to mTOR inhibitors may be due to an effect of Snail on mTOR activity rather than the expression of 4EBP1.

Response to reviewers' comments

We sincerely thank the reviewers for carefully reading through our manuscript and providing valuable and constructive comments that have dramatically helped us to improve our manuscript and the presentation of our work. We have taken the comments from all reviewers seriously and revised our manuscript extensively. We believe that with our new data, the reversion has dramatically strengthened our study and addressed the reviewers' concerns. Below, we respond to the comments made by each reviewer.

Reviewer #1 (Remarks to the Author):

In this article the authors describe the regulation by Snail1 of 4E-BP1, cap-dependent translation inhibitor. The repressive activity of 4E-BP1 is abrogated by its phosphorylation by TORC1 kinase. The authors adequately demonstrate that Snail1 represses 4E-BP1 transcription and down-regulates its expression, inducing resistance to mTORC kinase inhibitors. These experiments are adequately designed and performed and the conclusions of the first part (regulation of 4E-BP1 by Snail1) are solid. However, since Snail1 also activates other proteins regulated by TORC such as Akt (see Vega et al, 2004 or Escriva et al 2008, op. cit) it is not clear to me that the increased resistance to TORC inhibitors detected in Snail1-expressing cells is due uniquely (or mainly) to the repression of 4E-BP-1 and not to the activation of Akt. The authors should perform additional experiments to study the different contribution of these two elements: 1)they should analyze Akt activity in cells treated with TORCKis, either expressing Snail1 or not; 2) they should determine TORCKis sensitivity in Snail1-expressing cells transfected with 4E-BP1; and 3) in Snail1-depleted cells they should down-regulate 4E-BP1 or over-express Akt and analyze the different contribution of both elements to TORCKis sensitivity. These experiments will greatly increase the general interest of the article.

Response: We thank Reviewer#1 for recognizing the solid data we have on Snail regulation of 4E-BP1 expression, and greatly appreciate the constructive comments and valuable suggestions from Reviewer#1. As suggested by Reviewer#1, we first determined whether Snail can activate AKT through repression of PTEN as reported by Vega *et al*¹ and Escriva *et al*². Consistent with the findings of Vega *et al* and Escriva *et al* who expressed Snail in the MDCK cell line in their studies^{1,2}, we reproduced the finding that expression of Snail in MDCK cells represses PTEN expression (Supplementary Fig. 5c,d). In addition, we investigated three cell lines expressing exogenous Snail (T47D, MCF7, HCT116), as shown in Fig. 2a. We found that Snail expression only slightly repressed PTEN expression, coincident with enhanced activation of AKT, in the T47D cells but not in other two cell lines (Supplementary Fig. 9a,b). Furthermore, treatment with mTORkis AZD8055 or INK128 at the same concentration (100 nM) used for the cell growth assay effectively inhibited phosphorylation of mTOR substrates, including AKT, S6K and 4E-BP1 as well as the S6K substrate, S6 ribosomal protein, in both Snail-expressing or vector control T47D cells (Fig. 6d). However, the Snail-expressing T47D cells with decreased 4E-BP1 level (Figs. 2a and 6d) were still significantly insensitive to mTORkis compared with the vector control cells (Fig. 5c). Additionally, silencing Snail in HCT116, MDA-231 and SUM149 cells (Fig. 2c) did not affect PTEN expression and AKT/S6K phosphorylation (Supplementary Fig. 9c,d). It is important to note that

SUM149 cells have a naturally occurring PTEN loss associated with constitutive activation of AKT³; silencing Snail could still sensitize SUM149 cells to mTORkis (Figs. 2c and 5d), suggesting that the increased resistance to mTORkis by Snail expression is independent of AKT activation. This notion was supported by overexpression of AKT1 in Snail-depleted HCT116 cells. In this setting, AKT1 expression did not significantly prevent the cell growth inhibition, either with or without INK128 (Fig. 6e,f). The discordant Snail effect on PTEN expression and AKT activation between Escriva *et al* and our current study is probably associated with the differences in the type of cell and species used. Alternatively, this situation might be related to the lower ability of Snail to repress the PTEN promoter as suggested by Escriva *et al*.

To evaluate whether Snail-regulated 4E-BP1 is required for cell sensitivity to mTORkis, we first stably expressed exogenous 4E-BP1 to rescue the loss of 4E-BP1 in Snail-expressing MCF7 and T47D cells (Fig. 2a and Fig. 6a). Interestingly, we found that restoration of 4E-BP1 suppressed growth in Snail-expressing MCF7 or T47D cells, and that this suppression was enhanced by AZD8055 or INK128 (Fig. 6b). Next, as we showed in the original manuscript, silencing 4E-BP1 using shRNA to reverse the increased 4E-BP1 level induced by *Snail* knockout in HCT116 cells largely attenuated cell growth inhibition by *Snail* knockout alone or in combination with INK128 (Fig. 6c,g). Thus, these data reveal a critical role for 4E-BP1 in the modulation of cell sensitivity to mTORkis by Snail.

We have included these new data in the revised manuscript as Fig. 6a,b,d-f, Supplementary Fig. 5c,d and Supplementary Fig. 9a-d; described the results on pages 13-14; and discussed the data on pages 19-20.

Other comments.

1) It is accepted in the field that Snail1 overexpression in epithelial cells decreases cell proliferation as initially reported by Vega et al 2004, who indicated that Snail1 repressed cyclinD2 promoter. However, for the authors, Snail1 expression increases proliferation in MCF7 and T47D cell lines and do not alter cyclin D2, as mentioned in the Discussion. They have tried to explain this discrepancy indicating that it might be matter of cells specificity. I agree with them if we compare mesenchymal with epithelial cells but not when expressing Snail1 in epithelial cells; since 2004 many other groups have reproduced Vega's result in other cell lines. Is the effect observed by the authors also reproduced in MDCK cells? Is this a consequence of lower expression of Snail1 compared with that in Vega et al experiments? Does their Snail1 construct hold any mutation that modifies its capability to repress? The authors should try to perform some assays to solve this conflict, not merely indicating that it is a matter of cell line because the same argument might also apply to their results.

Response: We agree with the points raised by Reviewer#1. We apologize for not providing our Snail construct information in more detail. The human Snail construct was generated by my colleague, Binhua Zhou, who is an expert in Snail regulation and function related to EMT and breast cancer metastasis, and is now included as a co-author in the revised manuscript due to his contributions in providing several critical constructs, suggestions and discussion on this study. Human Snail was cloned into pLenti6.3 lentivirus expression vector (Invitrogen) and this construct was used to express Snail in a number of cell lines including MCF7 and T47D, and has been

reported by Dr. Zhou's group⁴⁻⁶. The Snail construct was verified by automated DNA sequencing, and no any mutations were found in Snail sequence. Moreover, the expression level of Snail and its ability to repress E-cadherin and induce EMT in multiple cell lines using this pLenti6.3 lentivirus expression vector is compatible with those obtained by using other expression vectors such as pCDNA3 and CMV-Tag2B as presented in Dr. Zhou's previous and recent studies⁴⁻¹¹. We have now added and cited our Snail construct information more clearly in the Results and Methods section on pages 5 and 22.

To address the question by Reviewer#1 whether the effect observed in our study also reproduced in MDCK cells, we generated a population pool of stable Snail-expressing MDCK cells using our pLenti6.3-human Snail construct as outlined for other cell lines used in this study. We found that expression of Snail in MDCK cells effectively repressed 4E-BP1 promoter activity and its expression (Supplementary Fig. 5), and that the mixed stable Snail-expressing MDCK cell population showed a significant increase in cell proliferation and G1/S progression and relieved the anti-proliferative effect of mTORkis (Supplementary Fig. 12b-d). These results are similar to those observed for MCF7 and T47D cell lines tested in this study. Consistent with the findings by Vega *et al*¹, we also observed the repression of cyclin D2 and induction of EMT by Snail expression in MDCK cells (Supplementary Fig. 5c,d and 12a). However, the effect of Snail on cyclin D2 expression was not observed in most of cell lines tested in this study, and most importantly, silencing Snail in the cell lines that overexpress Snail did not cause an increase in the levels of cyclin D2 (Supplementary Fig. 9a-d). Since Vega *et al*¹ used a single stable Snail-expressing MDCK cell clone for their study, the discordant Snail-effect on cell proliferation and expression of cyclin D2 observed in our study is probably associated with diverse effects in the cell population expressing different levels of Snail, and the type of cell or tissue and species used. It is worth noting that Escriva *et al*² also used single stable Snail-expressing MDCK cell clones in their study, and showed no significant difference in cell cycle progression between Snail-expressing and mock control cell clones when they were not irradiated (Figure 1A in Escriva *et al*'s report).

We have included these new data in the revised manuscript as Supplementary Figures 5, 9 and 12; described the results on pages 9 and 14; and discussed the data on pages 19-20.

2) The authors should include a more detailed characterization of the commercial 4E-BP1 antibody (Novus) they have used for the IHC analysis since it is different from the one used in WB that seems to be specific.

Response: We agree with Reviewer#1 that the use of a specific antibody is critical for the IHC analysis. In the original manuscript, we did use the 4E-BP1 antibody from Cell Signaling Technology (Cat #9644), which is the same antibody we used for WB. This antibody is highly specific for 4E-BP1, as confirmed in our previous studies¹²⁻¹⁴. We apologize for not providing greater detail about the 4E-BP1 antibody, which may have caused Reviewer#1 confusion. We have now included the catalog number of the antibody in the Methods section on page 29 for clarity.

3) Since they indicate that eIF4E/4E-BP1 binding is modified by Snail1, they should show this by a CoIP assay; a m7GTP pull down does not substitute this analysis.

Response: We greatly appreciate the constructive comment from Reviewer#1. The m7GTP Sepharose beads that mimic the 5'mRNA cap to bind eIF4E and precipitate eIF4E-interacting proteins are widely used for analysis of cap-binding proteins^{12,15-19}. However, we agree with Reviewer#1 that a direct co-IP assay of eIF4E-interacting proteins would strengthen our study. We performed the experiment suggested by Reviewer#1. Consistent with the findings observed in the m7GTP pull down, the co-IP assay demonstrated that Snail expression in T47D cells profoundly decreased the binding of 4E-BP1 to eIF4E obtained in the presence of either AZD8055 or INK128, whereas silencing Snail in HCT116 cells had the opposite effect, with an increased 4E-BP1-eIF4E interaction when cells were treated with these two mTORkis (Fig. 4d,e). Furthermore, the decreased 4E-BP1-eIF4E interaction in Snail-expressing T47D cells treated with mTORkis was accompanied by increased formation of eIF4F translation initiation complex as noted by increased levels of eIF4G and eIF4A bound to eIF4E (Fig. 4d). Conversely, the increased 4E-BP1-eIF4E interaction in Snail-depleted HCT116 cells treated with mTORkis was accompanied by a decreased formation of the eIF4F complex (Fig. 4e).

We have included these new data in the revised manuscript as Fig. 4d,e and described the results on page 11.

4) Other factors have been shown to mimic or extend Snail1 repression of E-Cadherin. It would be useful for the researchers working in this area to know if Slug, Zeb1/2 or Twist also modulate 4E-BP1 levels and repress this promoter.

Response: We agree with Reviewer#1 that it would be beneficial to know whether Slug, Zeb or Twist also modulate 4E-BP1 levels and repress this promoter. To address this point, we performed the experiment suggested by Reviewer#1, wherein we transiently expressed Slug, Zeb1 and Twist1 in T47D cells that lack expression of these transcriptional factors (Supplementary Fig. 6a). To compare the ability of these transcriptional factors to regulate 4E-BP1 expression with that obtained with Snail, Snail and their control expression vector were also transiently expressed in T47D cells. Similar to Snail expression, Slug expression strongly repressed 4E-BP1 promoter activity and its mRNA and protein expression, whereas expression of Zeb1 or Twist1 did not significantly inhibit 4E-BP1 expression but did markedly repress E-cadherin expression (Supplementary Fig. 6b-d). Importantly, knockdown of Slug in MDA-231 and MDA-157 breast cancer cell lines that express high levels of Slug, by either transient transfection with siRNAs or stable expression of specific shRNAs, provided results similar to the knockdown of Snail: a profound increase in 4E-BP1 expression at both the mRNA and protein levels (Supplementary Fig. 6a,e-j). Silencing Zeb1 provided a dramatic increase in E-cadherin expression but had no effect on 4E-BP1 expression in both cell lines (Supplementary Fig. 6e,f,i,j). Moreover, silencing Twist1 also showed no effect on 4E-BP1 expression but significantly upregulated E-cadherin expression in the Twist1-overexpressing MDA-157 cells (Supplementary Fig. 6a,i,j). These data suggest that the Snail family member, Slug, may also function as a strong transcriptional repressor of 4E-BP1.

We have included these new data in the revised manuscript as Supplementary Fig. 6 and described the results on pages 9-10.

5) *Snail1* is generally accepted as a short range transcriptional repressor. In 4E-BP1 promoter some of the E-boxes, putative binding sites for *Snail1*, are more than 1000 bp upstream from the transcription start site, what is surprising. The ChIPs presented for *Snail1* binding to the three boxes in Figure 3d should be quantified and the results presented as “percentage of input” to provide a better determination of the relative binding.

Response: We appreciate this very valuable comment from Reviewer#1. Based on the Ensembl genome browser (<http://www.ensembl.org/index.html>), we realized that the positions of the three putative Snail-binding E-boxes in human 4E-BP1 promoter were not accurate, and we have now revised the positions in the manuscript as in Fig. 3a and Supplementary Fig. 3a, which shows a shorter position at each E-box compared to that in the original manuscript. While Snail is generally accepted as a short range transcriptional repressor, the Snail target, cyclin D2, contains the Snail-binding sites that are more than 1000 bp upstream of its transcription start site¹.

As suggested by Reviewer#1, we quantified the Snail-binding capacity in the three E-boxes as shown in Fig. 3d and presented the results as relative Snail occupancy showing the percentage of input (Fig. 3e). We found that Snail could occupy all three E-box regions of the 4E-BP1 promoter (Fig. 3d), although relative Snail occupancy was greater at E-boxes 2 and 3 than E-box 1 (Fig. 3e). Furthermore, E-box mutation analyses indicated that all three E-boxes including the one at position -1148 upstream of 4E-BP1 transcription start site were required for and cooperated in the Snail-mediated 4E-BP1 repression (Fig. 3a,g).

We have included these new data in the revised manuscript as Fig. 3a,e and Supplementary Fig. 3a and described the results on page 7.

6) *The inhibitor of 4E-BP1 phosphorylation INK128 substantially decreases Snail1 levels (see Figure 7C) likely due to the inhibitory effect of 4E-BP1 on Snail1 expression. However, this down-regulation in Snail1 is not accompanied by a 4E-BP1 increase. This discrepancy on the model should be responded. It is possible that the levels of Snail1 bound to 4E-BP1 Promoter are not modified by INK128 since this pool of Snail1 is probably more stable than the DNA-unbound Snail1. This hypothesis should be determined by a ChIP experiment of Snail1 in the presence or absence of the inhibitor.*

Response: This is an excellent point. We greatly appreciate the insightful comment from Reviewer#1. The discrepancy shown in Fig. 7c may be due to the longer exposure of the western blot for 4E-BP1, as a blot using a short exposure showed that the overall 4E-BP1 levels in INK128-treated HCT116 Snail-WT tumor samples (lanes 4-6) were higher than those in the Snail-WT control tumor samples (lanes 1-3). However, we also note that there is a variation at the level of 4E-BP1 among those samples (see the figure below).

We did find that downregulation of Snail (~50% inhibition) by INK128 treatment in HCT116 Snail-WT cells was not accompanied by a significant increase in 4E-BP1 levels (lane 2 vs lane 1 in Fig. 6g; Fig. 6e in the original manuscript). To address this concern, we performed the ChIP experiment suggested by Reviewer#1. We found that the levels of Snail bound to the 4E-BP1 promoter were indeed not affected by INK128 treatment (Supplementary Fig. 10). Thus, these data may explain why downregulation of Snail to a certain level by INK128 is not sufficient to result in a significant change in the levels of 4E-BP1 (Fig. 6g). Our data suggest that substantial repression of Snail expression or activity by pharmacological inhibitors or genetic disruption is required to increase 4E-BP1 expression.

We have replaced the 4E-BP1 blot using a short exposure in Fig. 7c; added the new data in the revised manuscript as Supplementary Fig. 10; and described the results on page 14.

7) The authors have reported that Snail1 expression is dependent on 4E-BP1 and 4E-BP1 phosphorylation (Cai et al, Oncotarget 2014, ref 17). This inhibition should be included in the model depicted in Figure 8 and the two reciprocal effects of Snail1 and 4E-BP1 better discussed in the appropriate section.

Response: We appreciate this helpful and constructive comment from Reviewer#1. We revised the Fig. 8 and the corresponding figure legend, and discussed the two reciprocal effects of Snail and 4E-BP1 on page 18.

Reviewer #2 (Remarks to the Author):

In this manuscript entitled 'Snail determines the therapeutic response to mTOR kinase inhibitors by transcriptional repression of 4E-BP1' the authors identify the zinc finger transcription factor Snail as a novel transcriptional regulator of 4EBP1. 4EBP1 an important negative regulator of cap-dependent protein synthesis via its interaction with the cap binding protein, eukaryotic translation initiation factor 4E (eIF4E). Several key oncogenes and tumor suppressors converge on eIF4E or its regulatory partners in order to modulate cap-dependent protein synthesis in cancer cells. Therefore, therapeutic agents that target components of the protein synthesis apparatus hold tremendous promise as anti-cancer drugs and a large emphasis has been placed on the development of therapeutics or novel treatment strategies to target the protein synthesis machinery in cancer including the use of mTOR inhibitors.

Importantly, Wang and colleagues present findings to suggest that Snail levels may regulate the therapeutic response to mTOR inhibitors by repressing 4EBP1 using a number of different cancer cell lines and in vitro cell proliferation assays. The authors clearly demonstrate that Snail selectively downregulates 4EBP1 expression at the transcription level and not other 4EBP family members by binding to specific sequences in the promoter of 4EBP1. While these findings may be novel, it is not clear whether this mechanism of regulation of 4EBP1 by Snail is evolutionarily conservation and is not an artifact of the cell models used. The clinical significance of Wang et al., findings is that Snail expression and function may modulate the response to mTOR inhibitors that are currently used in clinical trials. However, the authors often overstate the effect of

modulating Snail on the efficacy of mTOR inhibitors in the cell lines tested and in xenografts, which is at best modest, and fail to provide mechanistic insights into how 4EBP1 regulates cancer cell sensitivity to mTOR inhibitors in an unbiased manner. Importantly, it has been previously been published that 4EBP1 levels modulate sensitivity to mTOR inhibitors. Furthermore, the clinical significance of this study is not well discussed or experimentally investigated. Therefore, in the current form these findings do not advance the current understanding of the importance or relevance of Snail transcriptional effects on 4EBP1 expression and activity in regards to cancer treatment and this reviewer believes the manuscript is not suitable for publication in Nature Communications. Below are some suggestions that the authors may find useful for submission to a more specialized journal.

Response: We greatly appreciate the constructive and valuable comments from Reviewer#2. We have addressed all of the points suggested below and revised our manuscript extensively. We believe that our new data has dramatically strengthened our study as presented in this revised manuscript.

We continue to believe that the findings in this paper, on uncovering a novel function of Snail in translational control of tumorigenesis via repression of 4E-BP1, are of critical importance for tumor biology and therapeutics. We demonstrate for the first time that Snail is a strong transcriptional repressor of 4E-BP1, which promotes cap-dependent mRNA translation and protein synthesis in polysomes. Second, we reveal Snail as an important determinant of cancer cell sensitivity to mTOR kinase inhibitors via 4E-BP1-mediated translational control of cell proliferation. Finally, we demonstrate that genetic and pharmacological inhibition of Snail activity restores 4E-BP1 expression and largely improves therapeutic efficacy of mTOR kinase inhibitors *in vivo* in tumors with low 4E-BP1 activity. Specifically, 1) our work suggests that Snail expression may serve as a predictive marker for mTOR-targeted therapies; 2) this work highlights the importance of co-targeting Snail and mTOR for the treatment of tumors in which Snail overexpression occurs with reduced 4E-BP1 expression; and 3) this therapeutic regime can be accomplished with drugs (as suggested by Reviewer#2) that are now in clinical development. Taken together, our study has important basic and translational implications.

We hope that the revised manuscript is now suitable for publication in *Nature Communications*.

Major points:

1. Snail is an evolutionarily conserved zinc finger transcription factor involved in mesoderm formation in vertebrates. In cancer, Snail has a well-documented role in binding sequence specific E box region of promoters to repressor transcription of factors modulating epithelial-mesenchymal transition (EMT) and cell survival. In this manuscript, Wang et al., identify 4EBP1 as a previously uncharacterized transcriptional target of Snail. While the findings demonstrating that Snail directly influences the transcription of 4EBP1 are convincing, there is a strong concern that these affects may be an artifact of the cell models used. The authors should therefore confirm their findings that 4EBP1 is a conserved transcriptional target of Snail in mouse and also in a more physiological system such as normal human fibroblasts.

Response: This is an insightful question and one that we also were curious to answer. Per Reviewer#2's suggestions, we first determined whether Snail also regulates 4E-BP1 expression in normal human fibroblasts. It is known that Snail is highly expressed in fibroblasts associated with loss of E-cadherin²⁰. We found that silencing Snail using siRNAs 48 h after transfection in two Snail-expressing normal human fetal lung fibroblasts (IMR-90 and TIG1) produced a marked increase in the transcript and protein levels of both 4E-BP1 and E-cadherin (Supplementary Fig. 2). Next, we searched for the Snail-binding E-box on the 4E-BP1 promoter from different species. Interestingly, we found that the human Snail-binding E-boxes (CAGGTG or CACCTG) and their surrounding sequences are highly conserved among primate species, chimpanzee and monkey (Supplementary Fig. 4a). The mouse and rat do not share the exact human Snail-binding E-box sequences on their 4E-BP1 promoters, but the CACTTG or CAAGTG core, which is a relatively strong Snail-binding motif²¹, was found on their 4E-BP1 promoters and is highly conserved between mouse and rat (Supplementary Fig. 4b). One E-box sequence (position -75/-70) was identified immediately upstream of the mouse 4E-BP1 transcriptional start site and two additional E-boxes were identified at positions -929/-924 and -940/-935 from the transcription start site (Supplementary Fig. 4b). We tested the ability of Snail to repress the isolated mouse 4E-BP1 promoter (position -1430/+80) by luciferase reporter assay. We found that expression of either human or mouse Snail could effectively repress mouse 4E-BP1 promoter activity as well as its expression at both the protein and mRNA levels in NMuMG mouse mammary epithelial cells and in 4T1 mouse mammary carcinoma cells (Supplementary Fig. 4c-e). Conversely, knockdown of mouse Snail expression in NIH3T3 fibroblasts markedly upregulated 4E-BP1 protein and mRNA expression (Supplementary Fig. 4f,g). As a positive control, mouse E-cadherin expression was also negatively regulated by Snail expression (Supplementary Fig. 4d-g). Similarly, the canine 4E-BP1 promoter also contains three putative Snail-binding E-boxes upstream of its transcription start site (Supplementary Fig. 5a). Ectopic expression of human Snail also profoundly suppressed 4E-BP1 promoter activity and its expression in Madin-Darby canine kidney (MDCK) cells (Supplementary Fig. 5b-d), a cell line that is widely used to study gene transcriptional regulation by Snail. Taken together, these results strongly suggest that 4E-BP1 is a conserved transcriptional target of Snail.

We have included these new data in the revised manuscript as Supplementary Figures 2, 4 and 5; and described the results on pages 6, 8 and 9.

2. Wang and colleagues present a number of figures (4-7) supporting a role of Snail modulation of 4EBP1 levels which in turn regulate cap-dependent translation and the therapeutic response of cancer cells to mTOR inhibitors. Several panels relating to figures 4-6 are redundant and would normally be considered supplementary data. Furthermore, at times, the authors overstate their findings on the effect of Snail expression on the efficacy of mTOR inhibitors, which is quite modest. For example, the authors' statement that 'Repression of 4E-BP1 expression by Snail renders cancer cell resistant to mTORkis' is misleading, as in Figure 5a-b the authors are comparing distinct cancer cell lines that harbor numerous genetic alterations and are from different tissue of origin. Moreover, in Figure 5C cells expressing exogenous Snail are sensitivity to mTOR inhibitors when compared to isogenic control cells. Therefore, it is recommended that Wang et al., revise these points and clearly modify the text so as to accurately reflect the data presented.

Response: We deeply appreciate the constructive comments and valuable suggestions from Reviewer#2. To make each figure more clear and concise as suggested by Reviewer#2, we reorganized figures 4-6 by moving redundant panels, such as Fig. 4e,f and Fig. 6a,b in the original manuscript to the supplemental information section, Fig. 6c in the original manuscript to Fig. 4c in the revised manuscript, and adding new data in Fig. 4 and 6 (see below and the response to Reviewer#1's comment 3 above) to strengthen our conclusions.

While we believe that our updated data further support our contention that Snail plays an important role in the modulation of cancer cell sensitivity via transcriptional repression of 4E-BP1, we agree with Reviewer#2 that we did not accurately describe and present the data in the text at times. We have now extensively revised the text in the abstract, results, discussion and figure legends to more accurately reflect the data we presented. For example, we have now changed the sentence on page 12 lines 284-285 to 'Repression of 4E-BP1 expression by Snail reduces the anti-proliferative effect of mTORkis.'

In Fig. 5c, our results showed that cells expressing exogenous Snail increased proliferation and were less sensitive to mTOR kinase inhibitors when compared to isogenic control cells. However, we do apologize for not describing the results more clearly in the original manuscript. We have now changed the sentence on page 12 lines 291-294 to 'Notably, ectopic expression of Snail in MCF7 and T47D cells significantly increased cell growth (Fig. 5c). Treatment with either AZD8055 or INK128 profoundly inhibited growth of these two cell lines, but this suppression was largely alleviated by Snail expression (Fig. 5c).'

3. In Figure 4, the authors demonstrate that 'Snail-mediated repression of 4E-BP1 promotes cap-dependent translation and relieves the inhibitory effect of mTORkis on translation'. The authors further demonstrate that 'Snail mitigates the anti-proliferative effect of mTORkis by translational control of cell cycle regulators such as cyclins D1 and D3'. However, Wang and colleagues have not taken into consideration that changes in gene expression at the translation level that are cap-independent may also modulate the sensitivity to mTOR inhibitors upon Snail repression and that targets other than Cyclin D1 and D3 may contribute to the observed phenotype. Therefore, the authors should undertake different approaches such as using reporter constructs to monitor changes in cap-independent translation.

4. Relating to the last point, it would be very important to determine whether a Snail-specific translational program exists during physiological conditions where Snail is known to play an important role, for example during EMT.

Response: We greatly appreciate the insightful comments from Reviewer#2. Since comment 4 is related to comment 3, we would like to address these two issues jointly.

In this study, we used a dual luciferase reporter system that monitors the ratio between cap-dependent and -independent translation initiation^{12,15,22}. We apologize for not describing the method and presenting the results more clearly in the results section of original manuscript. We found that expression of Snail in MCF7 and T47D cells significantly increased the cap-dependent translation rate but had no effect on IRES-driven cap-independent translation; additionally, Snail expression largely attenuated the cap-dependent translation inhibition induced by mTORkis,

AZD8055 and INK128 (Fig. 4a and Supplementary Fig. 7a). Conversely, knockdown of Snail in HCT116 and MDA-231 cells repressed cap-dependent but not cap-independent translation, and the inhibitory effect on cap-dependent translation was enhanced by AZD8055 or INK128 (Fig. 4b and Supplementary Fig. 7b). Most importantly, depletion of 4E-BP1 completely reversed the inhibitory effects of Snail knockdown or knockout alone, and in combination with INK128, on cap-dependent translation in HCT116 cells (Fig. 4c and Supplementary Fig. 7c,d). Thus, our data suggest that Snail activates cap-dependent translation and reduces the translational inhibition by mTORkis via repression of 4E-BP1.

This notion was further supported by several of our subsequent findings using different new approaches. First, we performed the cap-binding and Co-IP assays to determine whether Snail modulates the assembly of eIF4F translation initiation complex (eIF4E, eIF4A and eIF4G), a rate-limiting step in cap-dependent translation²³. We found that Snail-mediated repression of 4E-BP1 largely maintained the formation of eIF4F complex in response to mTORkis treatment (Fig. 4d,e). Second, we performed polysome analysis to determine whether Snail alters the composition of mRNAs associated with actively translated polysomes. We found that expression of Snail in MCF7 cells promotes polysome formation and almost completely prevented polysome inhibition by INK128 (Fig. 4f,g). By contrast, knockout of *Snail* in HCT116 cells decreased polysome formation and the addition of INK128 further impaired polysome assembly (Fig. 4h,i). Third, we measured nascent protein synthesis to determine whether Snail regulates protein production during active protein synthesis. We found that overexpression of Snail in T47D cells increased protein synthesis and largely prevented the inhibitory effect of INK128 on protein synthesis (Fig.4j), whereas *Snail* knockout in combination with INK128 in HCT116 cells resulted in a more marked suppression of protein synthesis (62%) than either *Snail* knockout or INK128 treatment alone (35%-38%) (Fig.4k). Collectively, our data indicate that Snail promotes recruitment of capping mRNAs to eIF4F complex for their active translation in polysomes, and that mTOR-regulated cap-dependent translation rate is altered by Snail-mediated repression of 4E-BP1.

Thus, our accumulated findings in the revised manuscript reveal Snail as a positive regulator of cap-dependent translation and an important determinant of cancer cell sensitivity to mTORkis via repression of 4E-BP1, suggesting that a Snail-specific translational program exists during tumorigenesis, EMT and metastasis. We determined that Snail/4E-BP1-mediated translational control of cyclins D1 and D3 may influence the anti-proliferative effect of mTORkis, but our results do not exclude the possibility that other Snail/4E-BP1-translationally regulated targets in polysomes may also modulate the cancer cell sensitivity to mTORkis, and this possibility requires further detail analysis and validation. We believe a detailed analysis and validation of Snail/4E-BP1-translationally regulated targets lie beyond the scope of this manuscript and hope to present relevant data in a subsequent publication detailing the Snail-initiated translational program that contributes to EMT and tumor progression. This manuscript focuses on the key role of Snail in tumorigenesis and response to mTORkis via repression of 4E-BP1.

We have included these new data in the revised manuscript as Fig. 4d-k and Supplementary Fig. 7a,b, described the results on pages 11-12; and discussed the data in detail on pages 17, 18 and 20.

5. While the authors' findings indicate that Snail expression may serve as a predictive marker for mTOR therapies, the authors do not provide experimental evidence of the

clinically significant of their findings using a more physiologically relevant system. For example, the authors should test whether inhibition of co-repressors of Snails transcriptional activity such as HDAC1 and HDAC2 may serve as efficient therapy in cancers with low 4EBP1 activity and in combination with mTOR inhibitors. Such experiments would further support the use of Snail as a putative modulator of 4EBP1 and mTOR inhibitors in cancer therapy.

Response: We really appreciate this illuminating comment from Reviewer#2. We have now performed the experiments as suggested. First, we analyzed the effects of vorinostat, a FDA-approved pan-HDAC inhibitor, on Snail-mediated repression of 4E-BP1 promoter activity. Similar to the data shown in Fig. 3b, ectopic expression of Snail in T47D cells reduced the activity of 4E-BP1 promoter to 30%, but this repression was totally eliminated by treatment with vorinostat (Fig. 7d). Furthermore, vorinostat treatment also significantly increased 4E-BP1 promoter activity in both HCT116 and MDA-231 cells that express endogenous Snail (Fig. 7e). The increased promoter activity of 4E-BP1 was corroborated by increased 4E-BP1 transcripts and protein levels after vorinostat treatment in HCT116 cells (Fig. 7f,g). Consistent with the previous study²⁴, vorinostat treatment also markedly increased the E-cadherin transcripts in HCT116 cells (Fig. 7f). These results strongly suggest that HDAC activity is also required for efficient suppression of the 4E-BP1 promoter. Next, we explored whether HDAC inhibition by vorinostat could mimic genetic depletion of Snail to improve the therapeutic effect of mTORkis. In the mouse model of HCT116 xenograft tumors with high Snail/low 4E-BP1 levels, treatment with either vorinostat (50 mg/kg) or INK128 (1.5 mg/kg) modestly decreased tumor size (Fig. 7h). However, vorinostat in combination with INK128 almost completely suppressed tumor growth. The body weight of mice exposed to different treatment regimens were monitored and revealed only marginal weight loss (e.g. < 10%) for mice treated with both drugs (Supplementary Fig. 11c). These findings indicate that HDAC inhibitors greatly enhance the antitumor effect of mTORkis, suggesting that combined inhibition of HDAC and mTOR may be an effective and actionable therapeutic strategy in cancers that overexpress Snail with low 4E-BP1 activity. Given that multiple clinical trials involving mTOR inhibitors (rapalogs, INK128) as monotherapy or in combination with HDAC inhibitors (vorinostat, panobinostat) are being tested in advanced cancers (e.g. NCI clinical trial identifiers NCT01058707, NCT02719691, NCT01087554, NCT00918333), our findings suggest that incorporating the analysis of Snail and 4E-BP1 expression in primary tumors may help to prospectively identify resistance to mTOR inhibitors in the clinic studies.

We have included these new data in the revised manuscript as Fig. 7d-h and Supplementary Fig. 11c, described the results on page 16; and discussed the data on pages 21.

Minor points:

1. On page 7 line 167, the sentence ‘Conversely, knockdown of Snail expression in HCT116 and MDA-231 cells repressed the translation activity, which was enhanced the inhibitory effect of AZD8055 or INK128 on translation (Fig. 4b).’ should be revised for clarity.

Response: We appreciate this constructive comment from Reviewer#2. We have revised this sentence to ‘Conversely, knockdown of Snail in HCT116 and MDA-231 cells repressed cap-dependent but not cap-independent translation, and the inhibitory effect on cap-dependent

translation was enhanced by AZD8055 or INK128 (Fig. 4b and Supplementary Fig. 7b).’ on page 10 lines 234-236.

2. In Figure 5, panel b the word ‘Sanil’ should be corrected to ‘Snail’.

Response: We appreciate for making this correction from Reviewer#2. We have changed it in Fig. 5b in the revised manuscript.

Reviewer #3 (Remarks to the Author):

The manuscript by the She group uncovers a regulation of 4EBP1 expression by the Snail transcription factor. In a number of cell lines and overexpression/knock-down studies, they convincingly show that Snail expression inversely correlates with 4EBP1 expression. They claim that this is due to a direct control of Snail on the 4EBP1 promoter and that this determines the sensitivity of cancer cells to treatments with mTOR inhibitors. These two points requires further experimental evidence. Otherwise the study is interesting and may be relevant for growth control and tumorigenesis.

1) It is unclear how fast and direct the effect of Snail on 4EBP1 transcription is. I could not easily find the information how long one has to wait to see an effect of Snail modulation on 4EBP1 expression.

Response: This is an insightful comment from Reviewer#3. We apologize for not providing greater detail in the time response to Snail regulation of 4E-BP1 expression. For the transient transfection experiment, we normally measure 4E-BP1 promoter activity and its mRNA and protein expression 36-48h after infection with Snail lentivirus or transfection with Snail siRNAs. We have now included this information in the Methods section on page 23 and the corresponding figure legends for clarity.

2) The major drawback of the study is that the authors did not measure mTOR activity after Snail overexpression and knockdown. In figure 6A and 6B, it looks like Snail KO and rescue have major effects on 4EBP1 band shift. This might indicate increased mTOR activity upon Snail gain-of-function. This should be directly tested by measuring the phosphorylation of other mTOR substrates, including S6K and Akt. It is possible that the sensitivity to mTOR inhibitors may be due to an effect of Snail on mTOR activity rather than the expression of 4EBP1.

Response: We appreciate the constructive and valuable comment from Reviewer#3. This comment is similar to the request by Reviewer#1. As described in our responses to Reviewer#1’s comment, we performed several experiments using different approaches including: 1) measurement of PTEN expression and the phosphorylation of AKT, S6K and S6, 2) expression of AKT1 or 4E-BP1, and 3) silencing 4E-BP1 in the relevant cellular models. Our additional new data (Fig. 6a-f, Supplementary Fig. 5c,d and Supplementary Fig. 9a-d) fully support our contention that Snail modulation of cancer cell sensitivity is mainly mediated at the level of 4E-BP1 but independent of AKT/S6K activation. We have described these new data in the revised manuscript on pages 13-14 and discussed the data on pages 19-20.

References

1. Vega, S. *et al.* Snail blocks the cell cycle and confers resistance to cell death. *Genes Dev.* **18**, 1131-1143 (2004).
2. Escriva, M. *et al.* Repression of PTEN phosphatase by Snail1 transcriptional factor during gamma radiation-induced apoptosis. *Mol. Cell Biol.* **28**, 1528-1540 (2008).
3. Saal, L.H. *et al.* Recurrent gross mutations of the PTEN tumor suppressor gene in breast cancers with deficient DSB repair. *Nat. Genet.* **40**, 102-107 (2008).
4. Dong, C. *et al.* Loss of FBP1 by Snail-mediated repression provides metabolic advantages in basal-like breast cancer. *Cancer Cell* **23**, 316-331 (2013).
5. Dong, C. *et al.* Interaction with Suv39H1 is critical for Snail-mediated E-cadherin repression in breast cancer. *Oncogene* **32**, 1351-1362 (2013).
6. Wang, Y., Liu, J., Ying, X., Lin, P.C. & Zhou, B.P. Twist-mediated Epithelial-mesenchymal Transition Promotes Breast Tumor Cell Invasion via Inhibition of Hippo Pathway. *Sci. Rep.* **6**, 24606 (2016).
7. Zhou, B.P. *et al.* Dual regulation of Snail by GSK-3beta-mediated phosphorylation in control of epithelial-mesenchymal transition. *Nat. Cell Biol.* **6**, 931-940 (2004).
8. Wu, Y. *et al.* Stabilization of snail by NF-kappaB is required for inflammation-induced cell migration and invasion. *Cancer Cell* **15**, 416-428 (2009).
9. Lin, Y. *et al.* The SNAG domain of Snail1 functions as a molecular hook for recruiting lysine-specific demethylase 1. *EMBO J.* **29**, 1803-1816 (2010).
10. Dong, C. *et al.* G9a interacts with Snail and is critical for Snail-mediated E-cadherin repression in human breast cancer. *J. Clin. Invest.* **122**, 1469-1486 (2012).
11. Wu, Y. *et al.* Dub3 inhibition suppresses breast cancer invasion and metastasis by promoting Snail1 degradation. *Nat. Commun.* **8**, 14228 (2017).
12. She, Q.B. *et al.* 4E-BP1 is a key effector of the oncogenic activation of the AKT and ERK signaling pathways that integrates their function in tumors. *Cancer Cell* **18**, 39-51 (2010).
13. Ye, Q., Cai, W., Zheng, Y., Evers, B.M. & She, Q.B. ERK and AKT signaling cooperate to translationally regulate survivin expression for metastatic progression of colorectal cancer. *Oncogene* **33**, 1828-1839 (2014).
14. Cai, W., Ye, Q. & She, Q.B. Loss of 4E-BP1 function induces EMT and promotes cancer cell migration and invasion via cap-dependent translational activation of snail. *Oncotarget* **5**, 6015-6027 (2014).
15. Roux, P.P. *et al.* RAS/ERK signaling promotes site-specific ribosomal protein S6 phosphorylation via RSK and stimulates cap-dependent translation. *J. Biol. Chem.* **282**, 14056-14064 (2007).
16. Choo, A.Y., Yoon, S.O., Kim, S.G., Roux, P.P. & Blenis, J. Rapamycin differentially inhibits S6Ks and 4E-BP1 to mediate cell-type-specific repression of mRNA translation. *Proc. Natl. Acad. Sci. USA.* **105**, 17414-17419 (2008).
17. Dowling, R.J. *et al.* mTORC1-mediated cell proliferation, but not cell growth, controlled by the 4E-BPs. *Science* **328**, 1172-1176 (2010).
18. Hsieh, A.C. *et al.* The translational landscape of mTOR signalling steers cancer initiation and metastasis. *Nature* **485**, 55-61 (2012).
19. Thoreen, C.C. *et al.* A unifying model for mTORC1-mediated regulation of mRNA translation. *Nature* **485**, 109-113 (2012).
20. Franci, C. *et al.* Expression of Snail protein in tumor-stroma interface. *Oncogene* **25**, 5134-5144 (2006).
21. Mauhin, V., Lutz, Y., Dennefeld, C. & Alberga, A. Definition of the DNA-binding site repertoire for the Drosophila transcription factor SNAIL. *Nucleic Acids Res.* **21**, 3951-3957 (1993).
22. Tsukumo, Y., Alain, T., Fonseca, B.D., Nadon, R. & Sonenberg, N. Translation control during prolonged mTORC1 inhibition mediated by 4E-BP3. *Nat. Commun.* **7**, 11776 (2016).

23. Sonenberg, N. & Hinnebusch, A.G. Regulation of translation initiation in eukaryotes: mechanisms and biological targets. *Cell* **136**, 731-745 (2009).
24. Peinado, H., Ballestar, E., Esteller, M. & Cano, A. Snail mediates E-cadherin repression by the recruitment of the Sin3A/histone deacetylase 1 (HDAC1)/HDAC2 complex. *Mol. Cell Biol.* **24**, 306-319 (2004).

Reviewers' comments:

Reviewer #1 (Remarks to the Author):

The group has performed an extensive and exhaustive work to respond to all the concerns that I expressed. Almost all of them have been adequately solved; however, an important question still remains. I still do not understand how Snail1 ectopic expression does not cause a retard in cell growth; this response has been widely observed not only in MDCK but in other cell lines too and by many different authors including myself. This effect is associated to an EMT, a conversion that the authors detect. I think that, for the sake of clarity and to avoid confusion in the field, the authors should explore this issue a little further. For instance, I am afraid that since they are working with cell populations, the cells with high expression of Snail1 are lost and they are left just with those that express less; maybe in those cells Snail1 levels are not enough to repress cell proliferation. In order to explore this possibility that it would be convenient to carry out a dose-response transfection of Snail1 in MDCK cells and determine proliferation, apoptosis resistance and E-cadherin, 4E-BP1, PTEN and CycD2 mRNA, to verify the relationship between these parameters and Snail1 levels.

Reviewer #2 (Remarks to the Author):

The authors satisfactorily addressed all of the comments of this reviewer.

Reviewer #3 (Remarks to the Author):

the authors addressed the issues previously raised

Response to reviewers' comments

We sincerely thank the reviewers for carefully reading through our manuscript and providing valuable and constructive comments that have helped us to dramatically improve our manuscript and the presentation of our work. Reviewer #2 and Reviewer #3 have no more concerns. Below, we respond to the comments made by Reviewer #1.

Reviewer #1 (Remarks to the Author):

The group has performed an extensive and exhaustive work to respond all the concerns that I expressed. Almost of all of them have been adequately solved; however, an important question still remains. I still do not understand how Snail1 ectopic expression does not cause a retard in cell growth; this response has been widely observed not only in MDCK but in other cell lines too and by many different authors included myself. This effect is associated to an EMT, a conversion that the authors detect. I think that, for the sake of clarity and to avoid confusion in the field, the authors should explore this issue a little further. For instance, I am afraid that since they are working with cell populations, the cells with high expression of Snail1 are lost and they are left just with those that express less; maybe in those cells Snail1 levels are not enough to repress cell proliferation. In order to explore this possibility that it would be convenient to carry out a dose-response transfection of Snail1 in MDCK cells and determine proliferation, apoptosis resistance and E-cadh, 4E-BP1, PTEN and CycD2 mRNA, to verify the relationship between these parameters and Snail1 levels.

Response: We greatly appreciate the constructive and valuable comments from Reviewer #1. Our findings demonstrating the positive role for Snail in cell proliferation via transcriptional repression of 4E-BP1 expression are consistent with several studies, as demonstrated with the Snail transgenic mouse skin model, the expression or disruption of Snail in the *Drosophila* ovarian model, and the PTEN-deficient glioblastoma cells with knockdown of Snail expression¹⁻⁴. We note that other groups reported opposite results, showing that ectopic expression of Snail in the canine normal epithelial MDCK cell line using a single cell clone undergoing EMT can block cell cycle progression and confer resistance to apoptosis by transcriptional repression of cyclin D2 and PTEN expression^{5,6}. In our current study, we utilized a population pool of the stable Snail-expressing cells to reduce biases, but we agreed with Reviewer #1 that Snail expression in the cell populations might not be high enough to induce a complete EMT and inhibit cell proliferation. As suggested by Reviewer #1, we generated single MDCK cell clones with different expression levels of Snail protein. Using these stable Snail-expressing clones, we found that expression of high level of Snail is required to maximally repress expression of E-cadherin, cyclin D2 and PTEN at both the protein and mRNA levels; induce a complete EMT; inhibit cell cycle progression and cell proliferation; and render cell resistance to serum deprivation-induced apoptosis (Supplementary Fig. 12a-g). These data confirmed the findings reported by Vega *et al*⁶ and Escriva *et al*⁶, suggesting that a complete EMT induction associated with inhibition of cell proliferation and resistance to apoptosis requires a higher level of Snail expression. However, it is noteworthy that 4E-BP1 expression in MDCK cells was effectively inhibited even at low to medium Snail expression levels. In addition, these cells showed a partial

EMT with no or weak inhibition of expression of E-cadherin, cyclin D2 and PTEN, but significantly increased cell proliferation and G1/S progression, and were resistant to apoptosis (Supplementary Fig. 12a-g). Similar results were observed in our previously revised manuscript using the mixed stable Snail-expressing MDCK cell populations that showed a moderate Snail expression level and relieved the anti-proliferative effect of mTORkis (Supplementary Fig. 12h-j in the current revised manuscript). Thus, the discordant Snail-effect on cell proliferation and survival is probably associated with diverse effects in the cell population expressing different levels of Snail with variable EMT status, and the type of cell or tissue and species used.

We have included these new data in the revised manuscript as Supplementary Fig. 12a-h; described and discussed the results highlighted in light grey on pages 18-19.

Reviewer #2 (Remarks to the Author):

The authors satisfactorily addressed all of the comments of this reviewer.

Reviewer #3 (Remarks to the Author):

The authors addressed the issues previously raised.

References

- 1 Jamora, C. *et al.* A signaling pathway involving TGF-beta2 and snail in hair follicle morphogenesis. *PLoS Biol.* **3**, e11 (2005).
- 2 Du, F. *et al.* Expression of snail in epidermal keratinocytes promotes cutaneous inflammation and hyperplasia conducive to tumor formation. *Cancer Res.* **70**, 10080-10089 (2010).
- 3 Tseng, C. Y., Kao, S. H. & Hsu, H. J. Snail controls proliferation of Drosophila ovarian epithelial follicle stem cells, independently of E-cadherin. *Dev. Biol.* **414**, 142-148 (2016).
- 4 Han, S. P. *et al.* SNAI1 is involved in the proliferation and migration of glioblastoma cells. *Cell Mol. Neurobiol.* **31**, 489-496 (2011).
- 5 Vega, S. *et al.* Snail blocks the cell cycle and confers resistance to cell death. *Genes Dev.* **18**, 1131-1143 (2004).
- 6 Escriva, M. *et al.* Repression of PTEN phosphatase by Snail1 transcriptional factor during gamma radiation-induced apoptosis. *Mol. Cell Biol.* **28**, 1528-1540 (2008).

REVIEWERS' COMMENTS:

Reviewer #1 (Remarks to the Author):

The authors have adequately responded to my last concern; I think the article is suitable for publication.

Response to reviewers' comments

Reviewer #1 (Remarks to the Author):

The authors have adequately responded to my last concern; I think the article is suitable for publication.

We thank for Reviewer #1 for supporting publication of our manuscript.